# IN–RIL: INTERLEAVED REINFORCEMENT AND IMITATION LEARNING FOR POLICY FINE-TUNING

## ABSTRACT

Imitation learning (IL) and reinforcement learning (RL) offer complementary strengths for robot learning, and yet each has severe limitations when used in isolation. Recent studies have proposed hybrid approaches to integrate IL with RL, but still face major challenges such as over-regularization and poor sample efficiency. Thus motivated, we develop IN–RIL, **IN**terleaved **R**einforcement learning and **I**mitation **L**earning, for policy fine-tuning, which periodically injects IL updates after multiple RL updates. In essence, IN–RIL leverages 'alternating optimization' to exploit the strengths of both IL and RL without overly constraining the policy learning, and hence can benefit from both the stability of IL and the expert-guided exploration of RL accordingly. Since IL and RL involve different optimization objectives, we devise gradient separation mechanisms to prevent their interference. Furthermore, our rigorous analysis sheds light on how interleaving IL with RL stabilizes learning and improves iteration efficiency. We conduct extensive experiments on Robomimic, FurnitureBench, and Gym, and demonstrate that IN–RIL, as a general plug-in compatible with various state-of-the-art RL algorithms, can improve RL sample efficiency, and mitigate performance collapse.

## 1 INTRODUCTION

Recent advances in robot learning have largely been driven by imitation learning (IL) and reinforcement learning (RL) (Black et al.; Chi et al., 2023; Fu et al., 2024; Wu et al., 2023). While these paradigms offer complementary strengths, each exhibits fundamental limitations when applied in isolation. Imitation learning approaches, such as behavioral cloning (Florence et al., 2022; Shafiullah et al., 2022), learn policies through supervised learning on expert demonstrations. While IL provides stable learning dynamics, it faces three critical challenges: costly expert demonstration collection (Zhao et al., 2024), limited generalization beyond the demonstration distribution, and vulnerability to compounding errors (Ankile et al., 2024; Rajeswaran et al., 2018). Even small deviations from demonstration distributions could accumulate and drastically degrade performance. RL approaches, in contrast, learn policies through environmental interaction to maximize

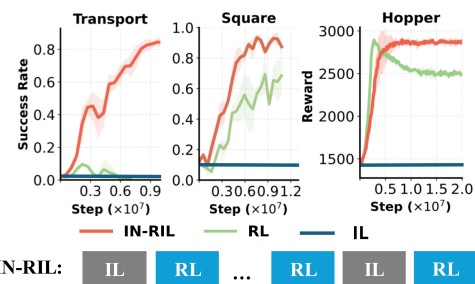

Figure 1: Performance of IL pre-training, RL-only fine-tuning, and IN–RIL (interleaved RL/IL) fine-tuning. IL pre-training saturates at low success rates; RL-only fine-tuning following IL-pre-training is unstable and sample-inefficient. By interleaving IL and RL updates, IN–RIL fine-tuning outperforms RL fine-tuning significantly.

accumulated rewards in a Markov Decision Process (MDP) (Sutton et al., 1999). RL enables active exploration beyond expert knowledge but often suffers from instability and poor sample efficiency (Hafner et al., 2023; Haarnoja et al., 2018). These problems are amplified in robotics tasks with sparse rewards and long horizons. For instance, as shown in Figure 1, IL alone yields poor performance due to the limited coverage of demonstrations, whereas RL-alone fine-tuning struggles to learn effectively. This has given rise to the following fundamental question on how to integrate IL and RL for robot learning:

*How to synergize the exploratory strengths of reinforcement learning with the stability of imitation learning for efficient policy fine-tuning?*

Existing hybrid approaches designed to combine IL and RL have made important progress (Florence et al., 2022; Nair et al., 2020; Song et al., 2022), but still face severe limitations. Notably, the approaches that initialize with IL pre-training followed by pure RL fine-tuning (Nakamoto et al., 2023; Nair et al., 2020) may suffer from policy collapse and instability as the learned policy drifts away from the IL initialization. Another line of works that inject demonstrations into replay buffers for off-policy updates (Ball et al., 2023; Song et al., 2022) require explicit reward annotations and complex sampling strategies, limiting their applicability to reward-sparse environments. The widely used strategy of imposing behavioral cloning (BC) regularization during RL fine-tuning (Rajeswaran et al., 2018; Haldar et al., 2023), adds a linearly weighted constraint to RL policy fine-tuning, treating IL and RL as competing objectives that must be balanced. This linear combination approach often leads to over-regularization or instability: when the weight favors IL, RL exploration is overly constrained; when it favors RL, the policy drifts away from stable IL guidance.

To address these limitations, we propose IN–RIL (INterleaved Reinforcement and Imitation Learning), a novel non-linear approach that alternates between IL and RL updates rather than linearly combining them. IN–RIL periodically interleaves IL updates after every few RL steps, allowing both optimization processes to progress effectively within their respective regimes. Our key insight is outlined below.

**Key Insight.** IL and RL objectives create fundamentally different non-convex optimization landscapes with multiple local optima. When used alone, each approach may get trapped in suboptimal regions of its landscape. As illustrated in Figure 2, IL can get stuck in demonstration-constrained minima while RL gets trapped in low-reward local optima. Observe that the regions that are local minima for IL may not be local minima for RL, and vice versa. The interleaving mechanism allows both objectives to benefit from each other's updates: RL updates help IL escape local minima in its loss landscape, while IL updates guide RL escape low-reward regions. This phenomenon is corroborated by our experiments where IL losses may experience a "double descent" (as illustrated in Section 3.3) and begin decreasing again despite the policy has converged during IL pre-training.

We caution that naively interleaving IL and RL updates introduces a new challenge: the different optimization landscapes can lead to destructive interference between gradient updates, potentially undermining the benefits of alternating optimization. This is because that the gradients from IL (which aim to match demonstrations) and RL (which aim to maximize rewards) may point in conflicting directions, causing oscillatory behavior or even preventing convergence. To address this challenge, we devise gradient separation mechanisms that effectively combine learning signals while preventing conflicts between these different objectives. In particular, we have developed two implementation approaches: (1) *gradient surgery* (Sener & Koltun, 2018; Quinton & Rey, 2024), which mitigates interference through gradient projection techniques; and (2) *network separation*, which isolates RL gradients in a residual policy while the base policy continues to leverage IL. Both methods effectively separate IL and RL gradient updates in different subspaces to prevent destructive interactions. It is worth noting that IN–RIL is algorithm-agnostic and can serve as a plug-in to existing RL algorithms, as demonstrated through our integration with state-of-the-art RL methods including DPPO (Ren et al., 2024), IDQL (Hansen-Estruch et al., 2023), residual PPO (Ankile et al., 2024; Yuan et al., 2024), covering both on-policy and off-policy approaches.

**Summary of Contributions.** In summary, our work makes the following contributions:

- **IN–RIL.** a fine-tuning approach that periodically inserts one IL iteration after every few RL iterations. IN–RIL synergizes the stability of IL using expert demonstrations with the exploration capabilities of RL throughout the fine-tuning process, and mitigates the instability and poor sample-efficiency of RL.
- **Gradient Separation Mechanisms.** Since IL and RL involve different optimization objectives, we devise two gradient separation mechanisms to prevent their interference in alternating optimization: gradient surgery using projection techniques and network separation via residual architectures. These mechanisms ensure that IL and RL gradients operate in separate subspaces while preserving the benefits of both objectives.
- **Analytic Foundation.** We provide theoretical analysis to shed light on why IN–RIL outperforms existing approaches by characterizing how alternating updates help each objective escape local

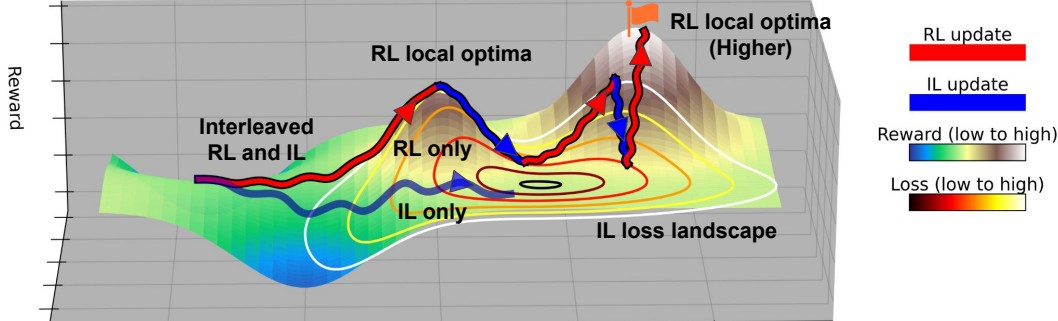

Figure 2: Optimization landscapes for IN–RIL. The IL loss landscape, represented by the 3D surface topology and its corresponding contour curves (where each contour connects points of equal IL loss value); and the landscape of RL rewards, represented by the color gradient mapped onto the surface (where the blue-to-white spectrum indicates low-to-high reward values as shown in the legend). IL updates drive the policy toward regions with lower losses, while RL updates steer toward higher rewards. Both optimization processes are stochastic and non-convex with multiple local optima. When using either RL or IL alone, training often converges to suboptimal solutions (as shown in the "IL only" and "RL only" trajectories). In contrast, IN–RIL enables RL and IL to help escape local optima: periodic IL updates help RL escape lower-reward regions toward higher-reward neighborhoods, while RL updates help IL traverse between different local minima in the loss landscape, as shown in our experiments in Section 3.3.

optima in their respective landscapes. Our analysis offers insights into optimal interleaving ratios and identifies the conditions under which mutual assistance between IL and RL is most effective.

- **Algorithm-Agnostic Design with Comprehensive Validation.** We demonstrate IN–RIL's effectiveness as a general plug-in compatible with state-of-the-art RL algorithms, including on-policy methods (DPPO, residual PPO) and off-policy approaches (IDQL). Through extensive experiments on challenging robotics tasks in FurnitureBench (Heo et al., 2023), Robomimic (Mandlekar et al., 2021), and OpenAI Gym (Brockman et al., 2016), we show substantial performance improvement on tasks with varying horizons and reward sparsity.

(**Related Work**) **Robotics Policy Learning and Fine-Tuning.** IL and RL each offer distinct advantages in policy learning: IL provides stable learning from expert demonstrations but struggles with distribution shift and demands costly data collection (Rajeswaran et al., 2018; Zhao et al., 2024), whereas RL enables exploration and generalization at the expense of high sample complexity, especially in long-horizon, sparse-reward settings (Song et al., 2022; Gupta et al., 2019). A common workaround is a two-stage pipeline—IL pre-training followed by RL fine-tuning (Ren et al., 2024; Ankile et al., 2024). IN–RIL moves beyond the two-stage paradigm, and shows that the data used for pre-training, even after pre-training plateaus, is still valuable in improving sample-efficiency and stability during RL fine-tuning. We attach more discussions to the Appendix.

**RL with Expert Demonstrations.** Recent works have explored leveraging offline data during RL. Several works (Haldar et al., 2023; Rajeswaran et al., 2018) introduce additional regularization terms to RL objectives to keep the policy close to expert behaviors, which often over-regularizes policy updates. Along a different line, very recent works on LLMs (Liu et al., 2025; He et al., 2025; Zhang et al., 2025) have proposed variants of such regularization for model fine-tuning, using a weighted sum of supervised fine-tuning (SFT) and RL objectives, Different from the regularization approaches which lead to one optimization landscape in the form of a weighted sum of IL and RL objectives, for the proposed IN–RIL based policy optimization, IL and RL updates keep their individual optimization landscapes and help each other to escape from their local minima. Another common approach is to add expert data with rewards to a replay buffer and perform off-policy updates during online learning (Nair et al., 2020; Song et al., 2022; Hu et al., 2023; Ball et al., 2023; Nakamoto et al., 2023). However, it can be infeasible to perform off-policy RL updates on expert demonstrations since reward annotations are not always available. Furthermore, sampling strategy is shown to be crucial for off-policy updates when there are both demonstration data and RL-collected data (Hu et al., 2023; Ball et al., 2023). In contrast, IN–RIL does not introduce explicit regularization terms which rely on delicate loss balancing, and can over-regularize the policy and damage performance. IN–RIL does not assume availability of rewards in IL data, or require sampling strategies to balance learning from offline and online data. Instead, it treats IL and RL as complementary optimization

processes and interleaves them during fine-tuning without modifying the RL algorithm itself. This makes IN–RIL broadly applicable to both on-policy and off-policy RL methods. IN–RIL follows standard assumptions on IL data coverage (as discussed in Section 2), and our analytics show that IN–RIL outperform RL even in the worst case where data coverage is low.

## 2 IN–RIL: INTERLEAVED RL AND IL FOR EFFICIENT POLICY FINE-TUNING

In this section, we provide a theoretical analysis of IN–RIL, aiming to answer two key questions: (1) What is the optimal interleaving ratio of RL updates to IL updates that balances learning stability and performance improvement, and (2) How much reduction in iteration complexity can be achieved by our proposed IN–RIL approach? We derive conditions under which IN–RIL achieves faster convergence to target performance levels.

**Markov Decision Process.** We consider a Markov Decision Process (MDP) defined by the tuple $\mathcal{M} = (\mathcal{S}, \mathcal{A}, P, r, \gamma, \rho_0)$, where $\mathcal{S}$ is the state space, $\mathcal{A}$ is the action space, $P : \mathcal{S} \times \mathcal{A} \times \mathcal{S} \to [0, 1]$ is the transition probability function, $r : \mathcal{S} \times \mathcal{A} \to \mathbb{R}$ is the reward function, $\gamma \in [0, 1)$ is the discount factor, and $\rho_0$ is the initial state distribution. A policy $\pi : S \to \Delta(A)$ maps states to probability distributions over actions. The action-value function, or Q-function, for a policy $\pi$ is defined as $Q^\pi(s, a) = \mathbb{E}_\pi \left[ \sum_{t=0}^\infty \gamma^t r(s_t, a_t) | s_0 = s, a_0 = a \right]$, representing the expected cumulative discounted reward when taking action $a$ in state $s$ and following policy $\pi$ thereafter. The objective in RL is to find a policy that maximizes the expected Q-value: $\mathbb{E}_{s \sim \rho_0, a \sim \pi(\cdot|s)}[Q^\pi(s, a)]$.

**Pre-Training.** We consider a parametric policy $\pi_\theta : \mathcal{S} \to \Delta(\mathcal{A})$ that maps states to distributions over actions. We employ a direct policy representation where $\pi_\theta(a|s)$ gives the probability (or probability density) of taking action $a$ in state $s$. This formulation allows for direct optimization through gradient-based methods while maintaining sufficient expressivity for complex robotic control tasks. During pre-training, we use behavior cloning to learn a policy that imitates expert demonstrations $\mathcal{D}_{\exp} = \{\tau_1, \tau_2, \dots, \tau_N\}$, where each trajectory $\tau_i = \{(s_1, a_1), \dots, (s_T, a_T)\}$ contains state-action pairs. The objective is to maximize the likelihood of expert actions given the corresponding states:

$$\mathcal{L}_{\mathrm{IL}}(\theta) = \mathbb{E}_{(s, a^*) \sim \mathcal{D}_{\exp}}[-\log \pi_\theta(a^*|s)], \tag{1}$$

where $a^*$ represents the expert action. This negative log-likelihood objective encourages the policy to assign high probability to actions demonstrated by experts in the same states. We then obtain a warm-start policy $\pi_0 = \arg\min_{\pi_\theta} \mathcal{L}_{\mathrm{IL}}(\theta)$ that serves as the initialization for subsequent fine-tuning. This pre-training approach allows the policy to capture the basic structure of the task before reinforcement learning is applied to further optimize performance. After obtaining a policy via imitation learning during pre-training, we proceed to the fine-tuning phase where we optimize the policy. In our analysis, we compare two distinct fine-tuning approaches, RL Fine-tuning and our proposed IN–RIL.

**RL Fine-tuning.** After pre-training, RL fine-tuning directly optimizes policy parameters to maximize the expected Q-value as defined earlier, through gradient updates of the form:

$$\theta_{t+1} = \theta_t - \alpha_{\mathrm{RL}} \nabla_\theta \mathcal{L}_{\mathrm{RL}}(\theta_t),$$

where $\alpha_{\mathrm{RL}}$ is the learning rate, and for convenience we define $\mathcal{L}_{\mathrm{RL}}(\theta) = -\mathbb{E}_{s \sim d^{\pi_\theta}}[Q^{\pi_\theta}(s, \pi_\theta(s))]$ as the loss function, which is the negative of the expected Q-value. This formulation directly connects to our optimization objective of maximizing $\mathbb{E}_{s \sim \rho_0, a \sim \pi(|s)}[Q^\pi(s, a)]$, but accounts for the evolving state distribution as the policy improves. While this approach aims to maximize the overall reward, it often suffers from instability and poor sample efficiency, particularly when fine-tuning complex models like diffusion policies.

**IN–RIL.** As illustrated in Figure 1, the proposed IN–RIL systematically alternates between IL and RL updates:

$$\theta_{t+\frac{1}{1+m(t)}} = \theta_t - \alpha_{\mathrm{IL}} \nabla_\theta \mathcal{L}_{\mathrm{IL}}(\theta_t),$$

$$\theta_{t+\frac{1+j}{1+m(t)}} = \theta_{t+\frac{j}{1+m(t)}} - \alpha_{\mathrm{RL}} \nabla_\theta \mathcal{L}_{\mathrm{RL}}(\theta_{t+\frac{j}{1+m(t)}}), \quad j \in \{1, \dots, m(t)\},$$

where $m(t)$ represents the iteration-dependent number of RL updates performed after each IL update. The IL updates help maintain the desirable behaviors from pre-training while providing regularization, and the RL updates improve performance on the target task.

Our analysis uses standard assumptions regarding the pre-training performance, data coverage, smoothness properties of the loss functions, and gradient estimation quality. Specifically, we assume that: (1) the initial policy obtained by pre-training results in a training loss within a bounded distance from the IL objective; (2) the expert demonstration dataset provides reasonably sufficient coverage of the relevant state space for the target task; (3) both the IL and RL objectives satisfy smoothness conditions; and (4) the stochastic gradient estimates for both objectives have bounded variance that decreases proportionally with batch size. The formal statements of these assumptions (Assumptions 2-5) and their implications are provided in Appendix A.

Next, we introduce the assumptions on the geometric relationship between the gradients of the IL and RL objectives in Assumption 1. In particular, we use the parameter $\rho(t)$ to capture the cosine similarity between these gradients, with positive values indicating opposing gradients and negative values indicating aligned gradients. Such assumption has been commonly used in multi-objective optimization (Sener & Koltun, 2018; Désidéri, 2012).

**Assumption 1** (Gradient Relationship). *In the fine-tuning regime, the gradients of IL and RL objectives exhibit the following relationship:*

$$\langle \nabla_\theta \mathcal{L}_{\mathrm{IL}}(\theta_t), \nabla_\theta \mathcal{L}_{\mathrm{RL}}(\theta_t) \rangle = -\rho(t) \|\nabla_\theta \mathcal{L}_{\mathrm{IL}}(\theta_t)\| \cdot \|\nabla_\theta \mathcal{L}_{\mathrm{RL}}(\theta_t)\|,$$

*where $\rho(t) \in [-1, 1]$ represents the time-varying relationship between gradients, with positive values indicating opposition (negative cosine similarity) and negative values indicating alignment (positive cosine similarity).*

Based on these assumptions, we establish the following key results on the optimal ratio of RL updates to IL updates in the proposed IN–RIL. This ratio is crucial for balancing the stability provided by IL updates with the performance improvements offered by RL updates.

**Theorem 1** (Optimal Interleaving Ratio). *Under Assumptions 1-5, at iteration $t$, the optimal ratio $m(t)$ for IN–RIL satisfies $m_{opt}(t) \geq 1$.*

Theorem 1 provides a principled formula for adapting the interleaving ratio throughout training based on current gradient information. The optimal ratio $m_{\mathrm{opt}}(t)$ increases when gradients strongly "oppose" each other ($\rho(t) < 0$) and decreases when they are more aligned ($\rho(t) > 0$), reflecting the intuition that more RL updates are needed to make progress when IL updates work against the RL objective. This result suggests that monitoring gradient alignment during training can help to determine the interleaving ratio for efficient optimization. Given this optimal ratio, we next quantify exactly how much more efficient IN–RIL can be compared to RL-only approaches. Denote $\Delta_{\mathrm{IL-RL}} = -\sum_{t=0}^{T-1} \frac{c_{\mathrm{IL}}\rho(t)}{L_{\mathrm{IL}}} \|\nabla \mathcal{L}_{\mathrm{IL}}(\theta_t)\| \cdot \|\nabla \mathcal{L}_{\mathrm{RL}}(\theta_t)\| - \frac{c_{\mathrm{IL}}^2 \sigma_{\mathrm{IL}}^2 T}{2L_{\mathrm{IL}} N_{\mathrm{IL}}}$. Then we have:

**Theorem 2** (Iteration Complexity of IN–RIL). *Under Assumptions 1-5, for a fixed computational budget of $T$ total updates, IN–RIL with $m > 1$ and $\Delta_{\mathrm{IL-RL}} > \frac{L_{\mathrm{RL}}(\mathcal{L}_{\mathrm{RL}}(\theta_0) - \mathcal{L}_{\mathrm{RL}}^*)}{m+1}$ requires fewer iterations to reach a target accuracy $\epsilon$ than RL-only fine-tuning, i.e., $\frac{T_{\text{RL-only}}}{T_{\text{IN–RIL}}} > 1$.*

Theorem 2 establishes the conditions under which IN–RIL achieves superior efficiency compared to RL-only fine-tuning. Specifically, when the regularization benefit $\Delta_{\mathrm{IL-RL}}$ exceeds the threshold $\frac{L_{\mathrm{RL}}(\mathcal{L}_{\mathrm{RL}}(\theta_0) - \mathcal{L}_{\mathrm{RL}}^*)}{m+1}$, IN–RIL requires fewer total updates to reach the same performance level. This threshold depends critically on the interleaving ratio $m$, with higher values of $m$ reducing the required regularization benefit for efficiency gain. Intuitively, this means that when the stabilizing effect of periodically revisiting the demonstration data is sufficiently strong, and the interleaving ratio is properly set, IN–RIL can achieve the same performance with fewer total updates. The result provides formal justification for the IN–RIL and offers practical guidance for setting the interleaving ratio based on task characteristics. In the next section, we show that this theoretical guarantee aligns with our empirical observations across multiple robotics tasks, where IN–RIL consistently demonstrates faster convergence and higher sample efficiency than pure RL approaches.

## 3 EXPERIMENTS

Based on the above analysis, we further conduct a comprehensive empirical evaluation to address two key questions: 1) What are the benefits of IN–RIL compared to RL fine-tuning and BC-regularized RL fine-tuning? 2) What is the impact of the interleaving ratio $m$ on the performance? To this end, we evaluate IN–RIL on three widely adopted benchmarks, including FurnitureBench (Heo et al., 2023), OpenAI Gym (Brockman et al., 2016), and Robomimic (Mandlekar et al., 2021). These benchmarks encompassing both locomotion and manipulation challenges with varying reward structures (sparse and dense) and time horizons (short and long).

**Robomimic (Mandlekar et al., 2021).** We evaluate IN–RIL on four Robomimic tasks: `Lift`, `Can`, `Square`, and `Transport`. Among these, `Square` and `Transport` are particularly challenging for RL agents (Ren et al., 2024). All tasks feature sparse rewards upon successful completion, with each task providing 300 demonstrations. For `Transport` and `Lift`, we specifically use noisy multi-human demonstration data to test robustness.

**FurnitureBench (Heo et al., 2023).** FurnitureBench featurs long-horizon, multi-stage manipulation tasks with sparse rewards. We include three assembly tasks: `One-Leg`, `Lamp`, and `Round-Table`. There are `Low` and `Med` randomness settings for state distributions. Each task includes 50 human demonstrations and provides sparse stage-completion rewards. We additionally incorporate two tasks from ResiP (Ankile et al., 2024): `Mug-Rack` and `Peg-in-Hole`.

**OpenAI Gym (Brockman et al., 2016).** To evaluate performance on dense-reward tasks, we include three classic locomotion benchmarks: `Hopper` (v2), `Walker2D` (v2), and `HalfCheetah` (v2). For these tasks, we utilize the medium-level imitation datasets from D4RL (Fu et al., 2020).

### 3.1 TRAINING

We evaluate IN–RIL with multiple policy parameterizations for pre-training, including diffusion policy (DP)(Chi et al., 2023) and Gaussian policy(Sutton et al., 1999), both of which are widely adopted in recent IL and RL literature (Ankile et al., 2024; Ren et al., 2024; Chi et al., 2023; Zhao et al., 2024). Particularly, DP has demonstrated superior performance across robotics tasks in both pre-training (Chi et al., 2023) and fine-tuning (Ren et al., 2024). We employ action chunking (Fu et al., 2024) to enhance temporal consistency. For fine-tuning, we select three state-of-the-art RL algorithms spanning both on-policy and off-policy approaches: 1) PPO (Schulman et al., 2017; Ankile et al., 2024; Yuan et al., 2024), a widely used on-policy algorithm; 2) DPPO (Ren et al., 2024), an on-policy, policy gradient-based RL algorithm; and 3) IDQL (Florence et al., 2022), an off-policy, Q-learning-based RL algorithm. IDQL and DPPO are both DP-based RL algorithms.

**Pre-Training.** Taking FurnitureBench as an example, we pre-train different policy parameterizations using 50 demonstrations with IL until convergence. As shown in Appendix Table 6, Gaussian policies without action chunking fails entirely on these challenging multi-stage sparse-reward tasks, while Gaussian policies with action chunking achieves limited success. DP demonstrates the strongest overall performance across all tasks. However, even DP pre-training remains sub-optimal, with 3 tasks showing below 5% success rates after loss plateaus, primarily due to limited dataset coverage.

**Fine-Tuning.** While DP yields the best pre-training performance, fine-tuning DP with conventional RL algorithms presents significant challenges and can fail (Ren et al., 2024; Yang et al., 2023). We consider two RL strategies: 1) *Full network fine-tuning*, where we use specialized DP-based RL algorithms to fine-tune DP; and 2) *Residual policy fine-tuning*, where we introduce an additional Gaussian policy as a residual policy on top of the pre-trained DP (base) policy. The residual policy is fine-tuned with conventional RL (PPO) (Schulman et al., 2017) while the base policy is updated with IL. The residual policy learns to adjust the base policy's actions at each time step. For each task, we fine-tune the pre-trained DP checkpoint with the highest success rate (or reward) using IN–RIL, and compare against RL-only fine-tuning. While our theory suggests an adaptive ratio $m(t)$, we use a constant value of $m$ throughout training for simplicity. Based on our results, values of $m$ between 5 and 15 work well across most tasks, balancing performance improvement with policy stability. We conduct a detailed ablation study on the impact of different $m$ values in Section 3.3.

**Separation of RL and IL gradients for IN–RIL.** RL and IL each operate within distinct optimization landscapes. Directly updating the same network with potentially conflicting objectives can

degrade policy performance (as demonstrated in our ablation study in Section 3.4). To address this, we introduce gradient separation mechanisms to prevent interference between RL and IL objectives: 1) *gradient surgery*, which projects each gradient onto the dual cone (Quinton & Rey, 2024), ensuring that updates benefit both individual objectives; and 2) *network separation*, which is naturally integrated with the residual RL fine-tuning strategy. This approach allocates IL gradients to the base policy while RL gradients update the residual policy, effectively mitigating interference.

### 3.2 IN–RIL PERFORMANCE

We demonstrate that IN–RIL can enhance the performance of state-of-the-art RL fine-tuning algorithms. For each benchmark, we select the best-performing RL algorithms according to recent literature: DPPO (Ren et al., 2024) and IDQL (Hansen-Estruch et al., 2023) for Robomimic and Gym tasks, and residual PPO (Ankile et al., 2024) for FurnitureBench. We also include other RL fine-tuning algorithms, AWC (Peng et al., 2019; Ren et al., 2024), and DIPO (Yang et al., 2023), in Table 1. We also compare IN–RIL with BC regularized RL fine-tuning (Rajeswaran et al., 2018) (denoted as "BC Reg" in the table), in which a BC loss is added to the RL objective.

**Robomimic and Gym fine-tuning results.** Figure 4 and Figure 3 show that IN–RIL consistently improves upon both IDQL and DPPO across most tasks. Notably, on the two most challenging Robomimic tasks, `Transport` and `Square` (Ren et al., 2024), IN–RIL substantially boosts performance of both IDQL and DPPO. The gains are especially prominent when combined with IDQL, where RL-only fine-tuning fails on `Transport` with 12% success rates, while IN–RIL successfully solves the task and achieves 88% success rates, as shown in Figure 3 and Table 1; on `Square`, IN–RIL improves IDQL by 22.5% in success rates; and reduces 62% environment steps needed for DPPO to converge in Figure 4. This highlights the crucial role of IL guidance for RL exploration. For Gym locomotion tasks, IN–RIL either matches or surpasses RL-only fine-tuning. In Figure 4, DPPO degrades after peaking on `Hopper`, while IN–RIL avoids this drop and ultimately surpasses

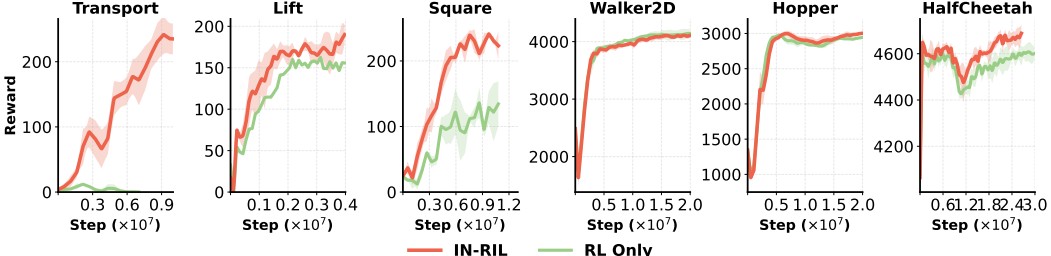

Figure 3: Comparing IN–RIL with RL fine-tuning on Robomimic and Gym using IDQL.

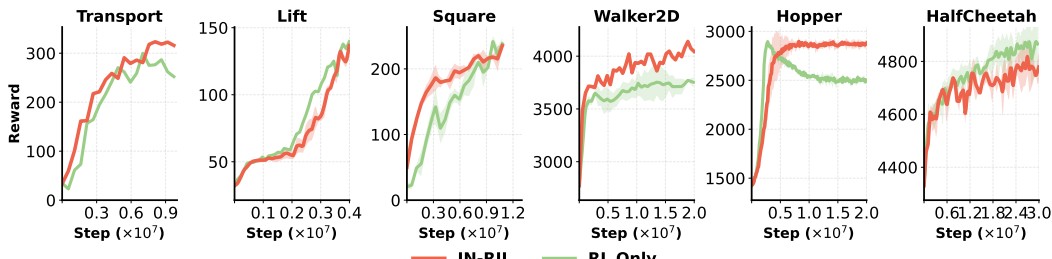

Figure 4: Comparing IN–RIL with RL fine-tuning on Robomimic and Gym using DPPO.

| Task | IN-RIL (DPPO) | DPPO | BC Reg (DPPO) | IN-RIL (IDQL) | IDQL | DIPO | AWR |
|---|---|---|---|---|---|---|---|
| Transport | *0.91±0.00* | 0.78±0.00 | 0.41 | **0.84±0.03** | 0.12±0.00 | 0.16 | 0.16 |
| Can | *1.00±0.00* | **1.00±0.00** | 0.96 | 0.98±0.00 | *1.00±0.00* | 0.94 | 0.65 |
| Lift | *1.00±0.00* | 0.93±0.05 | 0.98 | **0.99±0.00** | 0.99±0.00 | 0.97 | 0.99 |
| Square | *0.93±0.02* | 0.88±0.02 | 0.64 | **0.88±0.07** | 0.69±0.12 | 0.59 | 0.51 |
| Walker2D | **4045±0** | 3753±47 | 3457 | 4104±43 | **4143±79** | 3715 | *4250* |
| Hopper | **2901±31** | 2888±39 | 2896 | *3002±16* | 2943±24 | 2938 | 1427 |
| HalfCheetah | 4779±52 | *4866±149* | 4532 | **4688±39** | 4600±33 | 4644 | 4611 |

Table 1: Performance comparison for all fine-tuning methods on Robomimic (using success rates) and Gym tasks (using rewards). **Bold** values indicate the best in the DPPO group, or IDQL group. *Italic* values indicate the overall best across all methods.

**FurnitureBench fine-tuning results.** The multi-stage furniture assembly tasks with sparse rewards are particularly difficult for RL agents, especially when IL pre-training converges at low success rates, with 3 tasks below $5\%$, and only `One-Leg Low` over $30\%$, as demonstrated in Appendix Table 6. Meanwhile, IN–RIL significantly outperforms residual PPO across most tasks, as shown in Table 2, when consuming the same amount of environment steps. For the challenging `Lamp Low` task, RL-only fine-tuning frequently collapsed during training, while IN–RIL maintains stable learning dynamics across multiple runs. On `Round-Table Low`, where pre-training achieves only $5\%$ success rate, IN–RIL reaches $73\%$ success rate with approximately $10^8$ fewer environment interactions than RL-only fine-tuning with $25\%$ improvement in sample efficiency.

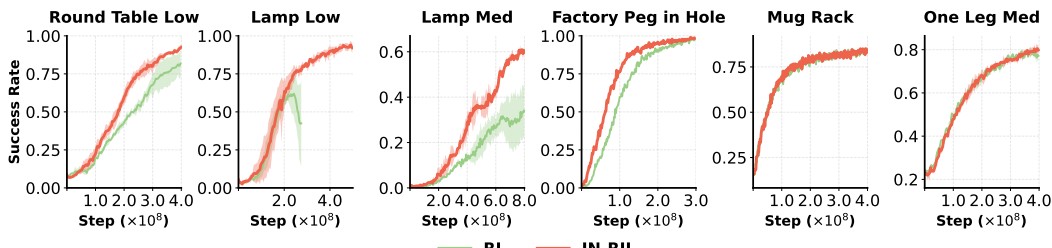

Figure 5: Comparing IN–RIL with RL fine-tuning on FurnitureBench using residual PPO.

| Task | IN-RIL (Residual PPO) | Residual PPO | DPPO | IDQL |
|---|---|---|---|---|
| Lamp (Low) | **0.92±0.04** | 0.42±0.28 | 0.85 | 0.11 |
| Lamp (Med) | **0.60±0.00** | 0.34±0.13 | 0.36 | 0.01 |
| Round-Table (Low) | **0.92±0.02** | 0.82±0.09 | 0.88 | 0.09 |
| One-Leg (Low) | 0.93±0.01 | **0.96±0.00** | 0.92 | 0.45 |
| One-Leg (Med) | **0.82±0.02** | 0.77±0.02 | 0.80 | 0.24 |

Table 2: Comparing IN–RIL with other RL fine-tuning algorithms on FurnitureBench. Bold values indicate the best of all.

## 3.3 Ablation Studies on Interleaving Ratio $m$

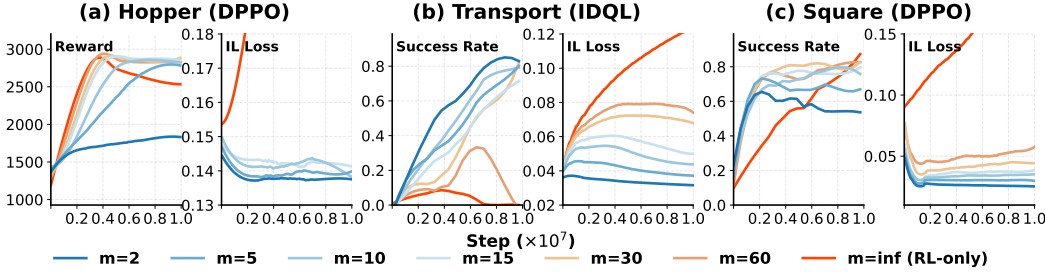

Figure 6: The impact of the interleaving period $m$ on IN–RIL RL performance (rewards), and IL performance (IL losses). We use 7 different values for $m$, and train the agent with all the values using $10^7$ environment steps. The figure shows how RL rewards and IL losses change with different $m$. The curves are smoothed using a Savitzky-Golay filter to better show the patterns.

Next, we investigate how the interleaving period $m$ affects the learning dynamics of IN–RIL by examining changes in both online performance metrics (RL rewards) and offline performance metrics (IL losses) under different values of $m$. For RL-only fine-tuning ($m = \infty$), we compute IL losses to monitor how well the policy maintains fidelity to demonstrations during fine-tuning, but without updating the policy based on these losses. We evaluate IN–RIL with seven different values of $m$. In particular, Figure 6 reveals several key insights about IN–RIL's behavior:

**Double Descent of IL Losses.** For RL-only fine-tuning ($m = \infty$), IL losses increase dramatically as RL drives the policy away from the pre-trained policy. In contrast, IN–RIL maintains controlled IL loss trajectories. Most remarkably, we observe that IL losses may experience a "double descent" phenomenon on some tasks—they begin decreasing again despite the pre-trained policy having fully converged. This empirically validates our hypothesis illustrated in Figure 2 that RL helps IL escape local minima, enabling discovery of superior policies that are inaccessible through IL alone.

**Enhanced Sample Efficiency.** Figure 6(c) demonstrates that IN–RIL dramatically improves the sample efficiency of DPPO, particularly during early fine-tuning. IN–RIL converges to high success rates within just $0.4 \times 10^7$ steps, while DPPO alone requires approximately $0.9 \times 10^7$ steps (1.25× more environment interactions) to achieve comparable performance.

**Improved Stability.** As shown in Figure 6(a), overly aggressive exploration in RL-only approaches can degrade performance after $0.4 \times 10^7$ steps. IN–RIL prevents this degradation across multiple interleaving ratios by maintaining IL losses within an appropriate range, effectively constraining exploration to promising regions of the policy space.

**Guided Exploration.** Figure 6(b) illustrates a critical advantage of IN–RIL: on challenging tasks where IDQL fine-tuning alone fails due to ungrounded exploration, IN–RIL successfully guides the agent toward task completion. By periodically refreshing the agent's memory of expert demonstrations through IL gradients, IN–RIL effectively structures exploration, enabling success on tasks that RL-only approaches cannot solve.

### 3.4 ABLATION OF GRADIENT SEPARATION.

When simultaneously leveraging IL and RL gradients to update policy networks, resolving potential interference between these distinct optimization objectives is crucial. When implementing gradient separation for IN–RIL with network separation, IL and RL gradients are naturally separated. In contrast, full-network fine-tuning, applies both gradients to the same network. To mitigate interference, we compute batch-wise IL and RL gradients, and apply gradient surgery updating the network. Figure 7 demonstrates that naive interleaving of IL and RL objectives without proper gradient management can significantly impair performance after IL updates, while separation enables successful learning.

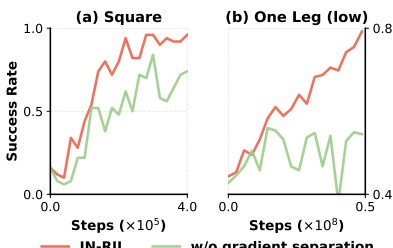

Figure 7: Impact of gradient separation on `Square` using IDQL and `One-Leg` (Low) using residual PPO.

## 4 CONCLUSION

We presented IN–RIL, a policy fine-tuning approach that interleaves IL and RL updates to leverage the stability of IL while promoting exploration and generalization through RL. To mitigate potential conflicts between these distinct learning signals, we introduced gradient separation mechanisms that prevent destructive interference during optimization, while retaining their benefits. Our theoretical analysis provides convergence guarantees and sample efficiency bounds, which are supported by empirical validation across three benchmark suites. As a modular and algorithm-agnostic plug-in, IN–RIL, when integrated with state-of-the-art RL fine-tuning algorithms, significantly improves performance across long- and short-horizon tasks with either sparse or dense rewards. Future directions include developing adaptive mechanisms to dynamically adjust the interleaving ratio based on gradient alignment, extending IN–RIL to domains beyond robotics, and exploring additional strategies to further enhance the synergy between IL and RL.

## 5 ETHICS STATEMENT

All experiments were conducted in simulation environments without involving human subjects, sensitive user data, or any form of personal information. Thus, there are no privacy, security, or human participant concerns. The datasets we use are publicly available benchmark datasets, and no proprietary or restricted data were employed. No conflicts of interest or external sponsorships influence the reported findings.

## 6 REPRODUCIBILITY STATEMENT

We take multiple steps to ensure reproducibility of our results. A detailed description of model architecture, training objectives, and algorithmic choices is provided in the main text. Hyperparameters and training configurations are reported in the Appendix. For theoretical derivations,

complete proofs and assumptions are included in the supplementary materials. To facilitate replication, we include anonymous source code with training scripts, evaluation pipelines, and configuration files as part of the supplementary material during review. All datasets used are publicly available.

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

## A APPENDIX

## B JUSTIFICATIONS ON THE ASSUMPTIONS

**Assumption 2** (Pretraining Performance). *The initial policy parameters $\theta_0$ obtained from pretraining satisfies $\mathcal{L}_{\mathrm{IL}}(\theta_0) - \mathcal{L}_{\mathrm{IL}}(\theta^*) \leq \epsilon_{\mathrm{IL}}$, where $\epsilon_{\mathrm{IL}} > 0$ is a constant and $\theta^*$ is the optimal solution for optimizing the IL objective.*

**Assumption 3** (Data Coverage). *The expert demonstration dataset $\mathcal{D}_{exp}$ provides sufficient coverage of the state space relevant for the target task. Specifically, there exists a constant $C_{coverage} > 0$ such that:*

$$\mathbb{E}_{s \sim \mu^*}[\min_{s' \in \mathcal{D}_{exp}} \|s - s'\|] \leq C_{coverage}$$

*where $\mu^*$ is the state distribution of the optimal policy for the target task.*

**Assumption 4** (Smoothness of Objectives). *Both the IL and RL objectives are L-smooth:*

$$\|\nabla_\theta \mathcal{L}_{\mathrm{IL}}(\theta) - \nabla_\theta \mathcal{L}_{\mathrm{IL}}(\theta')\| \leq L_{\mathrm{IL}}\|\theta - \theta'\|, \quad \forall \theta, \theta'$$
$$\|\nabla_\theta \mathcal{L}_{\mathrm{RL}}(\theta) - \nabla_\theta \mathcal{L}_{\mathrm{RL}}(\theta')\| \leq L_{\mathrm{RL}}\|\theta - \theta'\|, \quad \forall \theta, \theta'$$

**Assumption 5** (Bounded Variance). *The stochastic gradients have bounded variance:*

$$\mathbb{E}[\|\nabla_\theta \mathcal{L}_{\mathrm{IL}}(\theta) - \widehat{\nabla}_\theta \mathcal{L}_{\mathrm{IL}}(\theta)\|^2] \leq \frac{\sigma_{\mathrm{IL}}^2}{N_{\mathrm{IL}}}$$

$$\mathbb{E}[\|\nabla_\theta \mathcal{L}_{\mathrm{RL}}(\theta) - \widehat{\nabla}_\theta \mathcal{L}_{\mathrm{RL}}(\theta)\|^2] \leq \frac{\sigma_{\mathrm{RL}}^2}{N_{\mathrm{RL}}}$$

*where $\widehat{\nabla}$ represents the stochastic gradient estimate, and $N_{\mathrm{IL}}$ and $N_{\mathrm{RL}}$ are the batch sizes.*

We first provide the detailed justification on the assumptions used in Section 2.

**Assumption 1 (Near-Optimal IL Performance)** This assumption reflects the practical setting where we start from a pre-trained policy that already performs well on demonstration data. It's commonly used in transfer learning and foundation model literature where models are first trained on large datasets before task-specific adaptation Brown et al. (2020); Bommasani et al. (2021). The small constant $\epsilon_{IL}$ quantifies how close the initial policy is to optimal imitation performance, capturing the idea that while the model has learned a good behavioral prior, there's still room for improvement through reinforcement learning.

**Assumption 2 (Data Coverage)** The data coverage assumption ensures that the expert demonstrations provide adequate representation of the states relevant to the target task. This is a standard assumption in imitation learning Ross et al. (2011); Daumé et al. (2009) and reflects the intuition that learning can only occur for regions of the state space that have been demonstrated. The constant $C_{coverage}$ quantifies the maximum expected distance between a state from the optimal policy and its nearest neighbor in the demonstration dataset, with smaller values indicating better coverage.

**Assumption 3 (Smoothness of Objectives)** Smoothness is a standard assumption in optimization theory Nesterov (2004); Bottou et al. (2018)that ensures the gradient doesn't change too drastically between nearby points. This enables reliable gradient-based optimization and allows us to derive convergence rates. Practically, this assumption holds for most neural network architectures with commonly used activation functions when properly normalized, and is critical for establishing the descent lemma used in our analysis.

**Assumption 4 (Gradient Alignment)** This assumption characterizes the geometric relationship between the gradients of the IL and RL objectives. The parameter $\rho(t)$ captures the cosine similarity between these gradients, with positive values indicating opposing gradients and negative values indicating aligned gradients. Similar assumptions appear in multi-task learning literature Sener & Koltun (2018) and multi-objective optimization Désidéri (2012). This formulation allows us to analyze how the IL updates affect progress on the RL objective, which is crucial for determining the optimal interleaving strategy.

**Assumption 5 (Bounded Variance)** The bounded variance assumption is standard in stochastic optimization literature Robbins & Monro (1951); Bottou et al. (2018) and reflects the fact that stochastic gradient estimates contain noise due to mini-batch sampling. The variance terms $\sigma_{IL}^2$ and $\sigma_{RL}^2$ quantify this noise, with the variance decreasing as batch size increases. This assumption is necessary for establishing convergence rates in the presence of stochastic gradients and is satisfied in practice when using proper mini-batch sampling techniques.

Based on these assumptions, we first establish the following key results (proofs in the appendix). We begin our theoretical analysis by establishing convergence analysis for RL-only finetune and IN–RIL, respectively.

**Theorem 3** (Convergence of RL-Only Training). *Under Assumptions 2-5, with learning rate $\alpha_{\mathrm{RL}} = \frac{c_{\mathrm{RL}}}{L_{\mathrm{RL}}}$ for $c_{\mathrm{RL}} \in (0, 1)$, RL-only training for $T$ iterations achieves:*

$$\min_{0 \leq t < T} \mathbb{E}[\|\nabla \mathcal{L}_{\mathrm{RL}}(\theta_t)\|^2] \leq \frac{2L_{\mathrm{RL}}(\mathcal{L}_{\mathrm{RL}}(\theta_0) - \mathcal{L}_{\mathrm{RL}}^*)}{c_{\mathrm{RL}}(1 - \frac{c_{\mathrm{RL}}}{2})T} + \frac{c_{\mathrm{RL}}\sigma_{\mathrm{RL}}^2}{(1 - \frac{c_{\mathrm{RL}}}{2})N_{\mathrm{RL}}}$$

**Theorem 4** (Convergence with IN–RIL). *Under Assumptions 2-5, with learning rates $\alpha_{\mathrm{IL}} = \frac{c_{\mathrm{IL}}}{L_{\mathrm{IL}}}$ and $\alpha_{\mathrm{RL}} = \frac{c_{\mathrm{RL}}}{L_{\mathrm{RL}}}$ for $c_{\mathrm{IL}}, c_{\mathrm{RL}} \in (0, 1)$, interleaved 1:$m(t)$ training for $T$ cycles achieves:*

$$\min_{0 \leq t < T} \mathbb{E}[\|\nabla \mathcal{L}_{\mathrm{RL}}(\theta_t)\|^2] \leq \frac{2(L_{\mathrm{RL}}(\mathcal{L}_{\mathrm{RL}}(\theta_0) - \mathcal{L}_{\mathrm{RL}}^*) - \Delta_{\mathrm{IL-RL}})}{c_{\mathrm{RL}}(1 - \frac{c_{\mathrm{RL}}}{2})\bar{m}T} + \frac{c_{\mathrm{RL}}\sigma_{\mathrm{RL}}^2}{(1 - \frac{c_{\mathrm{RL}}}{2})N_{\mathrm{RL}}}$$

*where $\bar{m} = \frac{1}{T} \sum_{t=0}^{T-1} m(t)$ is the average interleaving ratio, and $\Delta_{\mathrm{IL-RL}}$ represents the benefit from IL regularization, i.e., $\Delta_{\mathrm{IL-RL}} = -\sum_{t=0}^{T-1} \frac{c_{\mathrm{IL}}\rho(t)}{L_{\mathrm{IL}}} \|\nabla \mathcal{L}_{\mathrm{IL}}(\theta_t)\| \cdot \|\nabla \mathcal{L}_{\mathrm{RL}}(\theta_t)\| - \frac{c_{\mathrm{IL}}^2 \sigma_{\mathrm{IL}}^2 T}{2L_{\mathrm{IL}}N_{\mathrm{IL}}}$*

Theorem 3 establishes that with appropriate learning rates, RL-only finetuning achieves the standard $O(1/T)$ convergence rate for smooth objectives. Theorem 4 reveals that IN–RIL can achieve better convergence guarantees than RL-only finetuning through the regularization benefit term $\Delta_{\mathrm{IL-RL}}$. This term captures how IL updates can enhance RL performance, especially when gradient alignment is favorable ($\rho(t) < 0$). Having established the benefits of IN–RIL, we now derive the optimal ratio of RL updates to IL updates. This ratio is crucial for balancing the stability provided by IL updates with the performance improvements offered by RL updates.

## C  PROOF OF THEOREM 3

We first establish the following technical lemmas that will be used in the proof of the main theorems.

**Lemma 1** (Descent Lemma). *For a function $f$ with $L$-smoothness, we have:*

$$f(y) \leq f(x) + \langle \nabla f(x), y - x \rangle + \frac{L}{2}\|y - x\|^2$$

**Lemma 2** (Progress Bound for Gradient Descent). *For a function $f$ with $L$-smoothness and step size $\alpha = \frac{c}{L}$ where $c \in (0, 1)$, one step of gradient descent gives:*

$$f(x - \alpha \nabla f(x)) \leq f(x) - \frac{c(1 - \frac{c}{2})}{L}\|\nabla f(x)\|^2$$

**Lemma 3** (Error Bound for Stochastic Gradient Descent). *For a function $f$ with $L$-smoothness, step size $\alpha = \frac{c}{L}$ where $c \in (0, 1)$, and stochastic gradient $\widehat{\nabla} f(x)$ with bounded variance $\mathbb{E}[\|\nabla f(x) - \widehat{\nabla} f(x)\|^2] \leq \frac{\sigma^2}{N}$, one step of stochastic gradient descent gives:*

$$\mathbb{E}[f(x - \alpha \widehat{\nabla} f(x))] \leq f(x) - \frac{c(1 - \frac{c}{2})}{L}\|\nabla f(x)\|^2 + \frac{c^2\sigma^2}{2LN}$$

*Proof.* The RL-only update rule is:

$$\theta_{t+1} = \theta_t - \alpha_{\mathrm{RL}}\widehat{\nabla}_\theta \mathcal{L}_{\mathrm{RL}}(\theta_t)$$

Where $\widehat{\nabla}_\theta \mathcal{L}_{\mathrm{RL}}(\theta_t)$ is the stochastic gradient estimate. Applying Lemma 3 to the RL objective, with $\alpha_{\mathrm{RL}} = \frac{c_{\mathrm{RL}}}{L_{\mathrm{RL}}}$:

$$\mathbb{E}[\mathcal{L}_{\mathrm{RL}}(\theta_{t+1})] \leq \mathcal{L}_{\mathrm{RL}}(\theta_t) - \frac{c_{\mathrm{RL}}(1 - \frac{c_{\mathrm{RL}}}{2})}{L_{\mathrm{RL}}} \|\nabla \mathcal{L}_{\mathrm{RL}}(\theta_t)\|^2 + \frac{c_{\mathrm{RL}}^2 \sigma_{\mathrm{RL}}^2}{2 L_{\mathrm{RL}} N_{\mathrm{RL}}}$$

Rearranging:

$$\frac{c_{\mathrm{RL}}(1 - \frac{c_{\mathrm{RL}}}{2})}{L_{\mathrm{RL}}} \|\nabla \mathcal{L}_{\mathrm{RL}}(\theta_t)\|^2 \leq \mathcal{L}_{\mathrm{RL}}(\theta_t) - \mathbb{E}[\mathcal{L}_{\mathrm{RL}}(\theta_{t+1})] + \frac{c_{\mathrm{RL}}^2 \sigma_{\mathrm{RL}}^2}{2 L_{\mathrm{RL}} N_{\mathrm{RL}}}$$

Summing from $t = 0$ to $T - 1$:

$$\frac{c_{\mathrm{RL}}(1 - \frac{c_{\mathrm{RL}}}{2})}{L_{\mathrm{RL}}} \sum_{t=0}^{T-1} \|\nabla \mathcal{L}_{\mathrm{RL}}(\theta_t)\|^2 \leq \mathcal{L}_{\mathrm{RL}}(\theta_0) - \mathbb{E}[\mathcal{L}_{\mathrm{RL}}(\theta_T)] + \frac{c_{\mathrm{RL}}^2 \sigma_{\mathrm{RL}}^2 T}{2 L_{\mathrm{RL}} N_{\mathrm{RL}}}$$

By Assumption 6, $\mathcal{L}_{\mathrm{RL}}(\theta_T) \geq \mathcal{L}_{\mathrm{RL}}^*$ (the optimal value), so:

$$\frac{c_{\mathrm{RL}}(1 - \frac{c_{\mathrm{RL}}}{2})}{L_{\mathrm{RL}}} \sum_{t=0}^{T-1} \|\nabla \mathcal{L}_{\mathrm{RL}}(\theta_t)\|^2 \leq \mathcal{L}_{\mathrm{RL}}(\theta_0) - \mathcal{L}_{\mathrm{RL}}^* + \frac{c_{\mathrm{RL}}^2 \sigma_{\mathrm{RL}}^2 T}{2 L_{\mathrm{RL}} N_{\mathrm{RL}}}$$

By the pigeonhole principle, there must exist at least one iteration $t^* \in \{0, 1, \ldots, T-1\}$ such that:

$$\|\nabla \mathcal{L}_{\mathrm{RL}}(\theta_{t^*})\|^2 \leq \frac{1}{T} \sum_{t=0}^{T-1} \|\nabla \mathcal{L}_{\mathrm{RL}}(\theta_t)\|^2$$

Therefore:

$$\min_{0 \leq t < T} \|\nabla \mathcal{L}_{\mathrm{RL}}(\theta_t)\|^2 \leq \frac{1}{T} \sum_{t=0}^{T-1} \|\nabla \mathcal{L}_{\mathrm{RL}}(\theta_t)\|^2 \leq \frac{L_{\mathrm{RL}}(\mathcal{L}_{\mathrm{RL}}(\theta_0) - \mathcal{L}_{\mathrm{RL}}^*)}{c_{\mathrm{RL}}(1 - \frac{c_{\mathrm{RL}}}{2})T} + \frac{c_{\mathrm{RL}}^2 \sigma_{\mathrm{RL}}^2}{2 c_{\mathrm{RL}}(1 - \frac{c_{\mathrm{RL}}}{2}) N_{\mathrm{RL}}}$$

Simplifying the second term:

$$\min_{0 \leq t < T} \|\nabla \mathcal{L}_{\mathrm{RL}}(\theta_t)\|^2 \leq \frac{L_{\mathrm{RL}}(\mathcal{L}_{\mathrm{RL}}(\theta_0) - \mathcal{L}_{\mathrm{RL}}^*)}{c_{\mathrm{RL}}(1 - \frac{c_{\mathrm{RL}}}{2})T} + \frac{c_{\mathrm{RL}} \sigma_{\mathrm{RL}}^2}{2(1 - \frac{c_{\mathrm{RL}}}{2}) N_{\mathrm{RL}}}$$

Taking expectation and adjusting the constant in the second term:

$$\min_{0 \leq t < T} \mathbb{E}[\|\nabla \mathcal{L}_{\mathrm{RL}}(\theta_t)\|^2] \leq \frac{2 L_{\mathrm{RL}}(\mathcal{L}_{\mathrm{RL}}(\theta_0) - \mathcal{L}_{\mathrm{RL}}^*)}{c_{\mathrm{RL}}(1 - \frac{c_{\mathrm{RL}}}{2})T} + \frac{c_{\mathrm{RL}} \sigma_{\mathrm{RL}}^2}{(1 - \frac{c_{\mathrm{RL}}}{2}) N_{\mathrm{RL}}}$$

For the IL performance bound, we use the $L_{\mathrm{IL}}$-smoothness of the IL objective (Assumption 3):

$$\mathcal{L}_{\mathrm{IL}}(\theta_T) - \mathcal{L}_{\mathrm{IL}}(\theta_0) \leq \langle \nabla \mathcal{L}_{\mathrm{IL}}(\theta_0), \theta_T - \theta_0 \rangle + \frac{L_{\mathrm{IL}}}{2} \|\theta_T - \theta_0\|^2$$

$$\leq \|\nabla \mathcal{L}_{\mathrm{IL}}(\theta_0)\| \cdot \|\theta_T - \theta_0\| + \frac{L_{\mathrm{IL}}}{2} \|\theta_T - \theta_0\|^2$$

From Assumption 1 (Near-Optimal IL Performance), the gradient $\|\nabla \mathcal{L}_{\mathrm{IL}}(\theta_0)\|$ is small. For simplicity, we can absorb this term into the quadratic term:

$$\mathcal{L}_{\mathrm{IL}}(\theta_T) - \mathcal{L}_{\mathrm{IL}}(\theta_0) \leq \frac{L_{\mathrm{IL}}}{2} \|\theta_T - \theta_0\|^2$$

Combining with Assumption 1, we have:

$$\mathcal{L}_{\mathrm{IL}}(\theta_T) - \mathcal{L}_{\mathrm{IL}}(\theta^*) = \mathcal{L}_{\mathrm{IL}}(\theta_T) - \mathcal{L}_{\mathrm{IL}}(\theta_0) + \mathcal{L}_{\mathrm{IL}}(\theta_0) - \mathcal{L}_{\mathrm{IL}}(\theta^*)$$

$$\leq \frac{L_{\mathrm{IL}}}{2} \|\theta_T - \theta_0\|^2 + \epsilon_{\mathrm{IL}}$$

This completes the proof. $\qquad \square$

# D    PROOF OF THEOREM 4

*Proof.* The interleaved training consists of cycles where each cycle has one IL update followed by $m(t)$ RL updates. Let $\theta_t$ denote the parameters at the beginning of cycle $t$, and $\theta_{t+\frac{j}{1+m(t)}}$ denote the parameters after the $j$-th update within cycle $t$.

First, let's analyze the IL update within cycle $t$:

$$\theta_{t+\frac{1}{1+m(t)}} = \theta_t - \alpha_{\mathrm{IL}}\widehat{\nabla}\mathcal{L}_{\mathrm{IL}}(\theta_t)$$

Applying Lemma 3 to the IL objective with $\alpha_{\mathrm{IL}} = \frac{c_{\mathrm{IL}}}{L_{\mathrm{IL}}}$:

$$\mathbb{E}[\mathcal{L}_{\mathrm{IL}}(\theta_{t+\frac{1}{1+m(t)}})] \le \mathcal{L}_{\mathrm{IL}}(\theta_t) - \frac{c_{\mathrm{IL}}(1-\frac{c_{\mathrm{IL}}}{2})}{L_{\mathrm{IL}}}\|\nabla\mathcal{L}_{\mathrm{IL}}(\theta_t)\|^2 + \frac{c_{\mathrm{IL}}^2\sigma_{\mathrm{IL}}^2}{2L_{\mathrm{IL}}N_{\mathrm{IL}}}$$

Now, let's analyze how this IL update affects the RL objective. Using the smoothness of the RL objective (Assumption 3):

$$\mathcal{L}_{\mathrm{RL}}(\theta_{t+\frac{1}{1+m(t)}}) \le \mathcal{L}_{\mathrm{RL}}(\theta_t) + \langle\nabla\mathcal{L}_{\mathrm{RL}}(\theta_t), \theta_{t+\frac{1}{1+m(t)}} - \theta_t\rangle + \frac{L_{\mathrm{RL}}}{2}\|\theta_{t+\frac{1}{1+m(t)}} - \theta_t\|^2$$

$$= \mathcal{L}_{\mathrm{RL}}(\theta_t) + \langle\nabla\mathcal{L}_{\mathrm{RL}}(\theta_t), -\alpha_{\mathrm{IL}}\widehat{\nabla}\mathcal{L}_{\mathrm{IL}}(\theta_t)\rangle + \frac{L_{\mathrm{RL}}\alpha_{\mathrm{IL}}^2}{2}\|\widehat{\nabla}\mathcal{L}_{\mathrm{IL}}(\theta_t)\|^2$$

Taking expectations and using the fact that $\mathbb{E}[\widehat{\nabla}\mathcal{L}_{\mathrm{IL}}(\theta_t)] = \nabla\mathcal{L}_{\mathrm{IL}}(\theta_t)$ (unbiased estimator):

$$\mathbb{E}[\mathcal{L}_{\mathrm{RL}}(\theta_{t+\frac{1}{1+m(t)}})] \le \mathcal{L}_{\mathrm{RL}}(\theta_t) - \alpha_{\mathrm{IL}}\langle\nabla\mathcal{L}_{\mathrm{RL}}(\theta_t), \nabla\mathcal{L}_{\mathrm{IL}}(\theta_t)\rangle + \frac{L_{\mathrm{RL}}\alpha_{\mathrm{IL}}^2}{2}\mathbb{E}[\|\widehat{\nabla}\mathcal{L}_{\mathrm{IL}}(\theta_t)\|^2]$$

Using Assumption 4 (Gradient align*ment):

$$\langle\nabla\mathcal{L}_{\mathrm{IL}}(\theta_t), \nabla\mathcal{L}_{\mathrm{RL}}(\theta_t)\rangle = -\rho(t)\|\nabla\mathcal{L}_{\mathrm{IL}}(\theta_t)\| \cdot \|\nabla\mathcal{L}_{\mathrm{RL}}(\theta_t)\|$$

And using Assumption 5 (Bounded Variance):

$$\mathbb{E}[\|\widehat{\nabla}\mathcal{L}_{\mathrm{IL}}(\theta_t)\|^2] \le \|\nabla\mathcal{L}_{\mathrm{IL}}(\theta_t)\|^2 + \frac{\sigma_{\mathrm{IL}}^2}{N_{\mathrm{IL}}}$$

We get:

$$\mathbb{E}[\mathcal{L}_{\mathrm{RL}}(\theta_{t+\frac{1}{1+m(t)}})] \le \mathcal{L}_{\mathrm{RL}}(\theta_t) + \alpha_{\mathrm{IL}}\rho(t)\|\nabla\mathcal{L}_{\mathrm{IL}}(\theta_t)\| \cdot \|\nabla\mathcal{L}_{\mathrm{RL}}(\theta_t)\|$$
$$+ \frac{L_{\mathrm{RL}}\alpha_{\mathrm{IL}}^2}{2}\left(\|\nabla\mathcal{L}_{\mathrm{IL}}(\theta_t)\|^2 + \frac{\sigma_{\mathrm{IL}}^2}{N_{\mathrm{IL}}}\right)$$

Substituting $\alpha_{\mathrm{IL}} = \frac{c_{\mathrm{IL}}}{L_{\mathrm{IL}}}$:

$$\mathbb{E}[\mathcal{L}_{\mathrm{RL}}(\theta_{t+\frac{1}{1+m(t)}})] \le \mathcal{L}_{\mathrm{RL}}(\theta_t) + \frac{c_{\mathrm{IL}}}{L_{\mathrm{IL}}}\rho(t)\|\nabla\mathcal{L}_{\mathrm{IL}}(\theta_t)\| \cdot \|\nabla\mathcal{L}_{\mathrm{RL}}(\theta_t)\|$$
$$+ \frac{L_{\mathrm{RL}}c_{\mathrm{IL}}^2}{2L_{\mathrm{IL}}^2}\left(\|\nabla\mathcal{L}_{\mathrm{IL}}(\theta_t)\|^2 + \frac{\sigma_{\mathrm{IL}}^2}{N_{\mathrm{IL}}}\right)$$

Now, let's analyze the $m(t)$ RL updates. For each RL update $j \in \{1, \ldots, m(t)\}$:

$$\theta_{t+\frac{1+j}{1+m(t)}} = \theta_{t+\frac{j}{1+m(t)}} - \alpha_{\mathrm{RL}}\widehat{\nabla}\mathcal{L}_{\mathrm{RL}}(\theta_{t+\frac{j}{1+m(t)}})$$

Applying Lemma 3 to each RL update, with $\alpha_{\mathrm{RL}} = \frac{c_{\mathrm{RL}}}{L_{\mathrm{RL}}}$:

$$\mathbb{E}[\mathcal{L}_{\mathrm{RL}}(\theta_{t+\frac{1+j}{1+m(t)}})] \le \mathcal{L}_{\mathrm{RL}}(\theta_{t+\frac{j}{1+m(t)}}) - \frac{c_{\mathrm{RL}}(1-\frac{c_{\mathrm{RL}}}{2})}{L_{\mathrm{RL}}}\|\nabla\mathcal{L}_{\mathrm{RL}}(\theta_{t+\frac{j}{1+m(t)}})\|^2$$
$$+ \frac{c_{\mathrm{RL}}^2\sigma_{\mathrm{RL}}^2}{2L_{\mathrm{RL}}N_{\mathrm{RL}}}$$

For simplicity of analysis, we can bound the gradient norms at intermediate steps using the gradient at the beginning of the cycle:

$$\|\nabla\mathcal{L}_{\mathrm{RL}}(\theta_{t+\frac{j}{1+m(t)}})\|^2 \geq (1-\delta)^2\|\nabla\mathcal{L}_{\mathrm{RL}}(\theta_t)\|^2$$

for some small $\delta > 0$ that depends on the learning rates and smoothness constants. This approximation is reasonable because the parameters don't change drastically within a cycle when using small learning rates.

With this approximation, we get:

$$\mathbb{E}[\mathcal{L}_{\mathrm{RL}}(\theta_{t+\frac{1+j}{1+m(t)}})] \leq \mathcal{L}_{\mathrm{RL}}(\theta_{t+\frac{j}{1+m(t)}}) - \frac{c_{\mathrm{RL}}(1-\frac{c_{\mathrm{RL}}}{2})(1-\delta)^2}{L_{\mathrm{RL}}}\|\nabla\mathcal{L}_{\mathrm{RL}}(\theta_t)\|^2$$
$$+ \frac{c_{\mathrm{RL}}^2\sigma_{\mathrm{RL}}^2}{2L_{\mathrm{RL}}N_{\mathrm{RL}}}$$

Applying this recursively for all $m(t)$ RL updates and combining with the effect of the IL update, we get:

$$\mathbb{E}[\mathcal{L}_{\mathrm{RL}}(\theta_{t+1})] \leq \mathcal{L}_{\mathrm{RL}}(\theta_t) + \frac{c_{\mathrm{IL}}}{L_{\mathrm{IL}}}\rho(t)\|\nabla\mathcal{L}_{\mathrm{IL}}(\theta_t)\| \cdot \|\nabla\mathcal{L}_{\mathrm{RL}}(\theta_t)\|$$
$$+ \frac{L_{\mathrm{RL}}c_{\mathrm{IL}}^2}{2L_{\mathrm{IL}}^2}\left(\|\nabla\mathcal{L}_{\mathrm{IL}}(\theta_t)\|^2 + \frac{\sigma_{\mathrm{IL}}^2}{N_{\mathrm{IL}}}\right)$$
$$- m(t)\frac{c_{\mathrm{RL}}(1-\frac{c_{\mathrm{RL}}}{2})(1-\delta)^2}{L_{\mathrm{RL}}}\|\nabla\mathcal{L}_{\mathrm{RL}}(\theta_t)\|^2 + m(t)\frac{c_{\mathrm{RL}}^2\sigma_{\mathrm{RL}}^2}{2L_{\mathrm{RL}}N_{\mathrm{RL}}}$$

For simplicity, we'll absorb $(1-\delta)^2$ into the constants. Rearranging:

$$m(t)\frac{c_{\mathrm{RL}}(1-\frac{c_{\mathrm{RL}}}{2})}{L_{\mathrm{RL}}}\|\nabla\mathcal{L}_{\mathrm{RL}}(\theta_t)\|^2 \leq \mathcal{L}_{\mathrm{RL}}(\theta_t) - \mathbb{E}[\mathcal{L}_{\mathrm{RL}}(\theta_{t+1})]$$
$$+ \frac{c_{\mathrm{IL}}}{L_{\mathrm{IL}}}\rho(t)\|\nabla\mathcal{L}_{\mathrm{IL}}(\theta_t)\| \cdot \|\nabla\mathcal{L}_{\mathrm{RL}}(\theta_t)\|$$
$$+ \frac{L_{\mathrm{RL}}c_{\mathrm{IL}}^2}{2L_{\mathrm{IL}}^2}\|\nabla\mathcal{L}_{\mathrm{IL}}(\theta_t)\|^2 + \frac{L_{\mathrm{RL}}c_{\mathrm{IL}}^2\sigma_{\mathrm{IL}}^2}{2L_{\mathrm{IL}}^2N_{\mathrm{IL}}}$$
$$+ m(t)\frac{c_{\mathrm{RL}}^2\sigma_{\mathrm{RL}}^2}{2L_{\mathrm{RL}}N_{\mathrm{RL}}}$$

Summing over $t = 0$ to $T-1$:

$$\sum_{t=0}^{T-1} m(t)\frac{c_{\mathrm{RL}}(1-\frac{c_{\mathrm{RL}}}{2})}{L_{\mathrm{RL}}}\|\nabla\mathcal{L}_{\mathrm{RL}}(\theta_t)\|^2 \leq \mathcal{L}_{\mathrm{RL}}(\theta_0) - \mathbb{E}[\mathcal{L}_{\mathrm{RL}}(\theta_T)]$$
$$+ \sum_{t=0}^{T-1} \frac{c_{\mathrm{IL}}}{L_{\mathrm{IL}}}\rho(t)\|\nabla\mathcal{L}_{\mathrm{IL}}(\theta_t)\| \cdot \|\nabla\mathcal{L}_{\mathrm{RL}}(\theta_t)\|$$
$$+ \sum_{t=0}^{T-1} \frac{L_{\mathrm{RL}}c_{\mathrm{IL}}^2}{2L_{\mathrm{IL}}^2}\|\nabla\mathcal{L}_{\mathrm{IL}}(\theta_t)\|^2 + T\frac{L_{\mathrm{RL}}c_{\mathrm{IL}}^2\sigma_{\mathrm{IL}}^2}{2L_{\mathrm{IL}}^2N_{\mathrm{IL}}}$$
$$+ \sum_{t=0}^{T-1} m(t)\frac{c_{\mathrm{RL}}^2\sigma_{\mathrm{RL}}^2}{2L_{\mathrm{RL}}N_{\mathrm{RL}}}$$

By Assumption 6, $\mathcal{L}_{\mathrm{RL}}(\theta_T) \geq \mathcal{L}_{\mathrm{RL}}^*$, so:

$$\sum_{t=0}^{T-1} m(t) \frac{c_{\mathrm{RL}}(1 - \frac{c_{\mathrm{RL}}}{2})}{L_{\mathrm{RL}}} \|\nabla \mathcal{L}_{\mathrm{RL}}(\theta_t)\|^2 \leq \mathcal{L}_{\mathrm{RL}}(\theta_0) - \mathcal{L}_{\mathrm{RL}}^*$$

$$+ \sum_{t=0}^{T-1} \frac{c_{\mathrm{IL}}}{L_{\mathrm{IL}}} \rho(t) \|\nabla \mathcal{L}_{\mathrm{IL}}(\theta_t)\| \cdot \|\nabla \mathcal{L}_{\mathrm{RL}}(\theta_t)\|$$

$$+ \sum_{t=0}^{T-1} \frac{L_{\mathrm{RL}} c_{\mathrm{IL}}^2}{2 L_{\mathrm{IL}}^2} \|\nabla \mathcal{L}_{\mathrm{IL}}(\theta_t)\|^2 + T \frac{L_{\mathrm{RL}} c_{\mathrm{IL}}^2 \sigma_{\mathrm{IL}}^2}{2 L_{\mathrm{IL}}^2 N_{\mathrm{IL}}}$$

$$+ \sum_{t=0}^{T-1} m(t) \frac{c_{\mathrm{RL}}^2 \sigma_{\mathrm{RL}}^2}{2 L_{\mathrm{RL}} N_{\mathrm{RL}}}$$

For the sum of IL gradient norms, we can use the IL update analysis. From our earlier bound on IL updates:

$$\sum_{t=0}^{T-1} \frac{c_{\mathrm{IL}}(1 - \frac{c_{\mathrm{IL}}}{2})}{L_{\mathrm{IL}}} \|\nabla \mathcal{L}_{\mathrm{IL}}(\theta_t)\|^2 \leq \mathcal{L}_{\mathrm{IL}}(\theta_0) - \mathbb{E}[\mathcal{L}_{\mathrm{IL}}(\theta_T)] + \frac{c_{\mathrm{IL}}^2 \sigma_{\mathrm{IL}}^2 T}{2 L_{\mathrm{IL}} N_{\mathrm{IL}}}$$

This gives us:

$$\sum_{t=0}^{T-1} \|\nabla \mathcal{L}_{\mathrm{IL}}(\theta_t)\|^2 \leq \frac{L_{\mathrm{IL}}(\mathcal{L}_{\mathrm{IL}}(\theta_0) - \mathcal{L}_{\mathrm{IL}}^*)}{c_{\mathrm{IL}}(1 - \frac{c_{\mathrm{IL}}}{2})} + \frac{c_{\mathrm{IL}} \sigma_{\mathrm{IL}}^2 T}{2(1 - \frac{c_{\mathrm{IL}}}{2}) N_{\mathrm{IL}}}$$

Substituting this bound and defining $\bar{m} = \frac{1}{T} \sum_{t=0}^{T-1} m(t)$ as the average interleaving ratio:

$$\bar{m} T \frac{c_{\mathrm{RL}}(1 - \frac{c_{\mathrm{RL}}}{2})}{L_{\mathrm{RL}}} \frac{1}{T} \sum_{t=0}^{T-1} \|\nabla \mathcal{L}_{\mathrm{RL}}(\theta_t)\|^2 \leq \mathcal{L}_{\mathrm{RL}}(\theta_0) - \mathcal{L}_{\mathrm{RL}}^*$$

$$+ \sum_{t=0}^{T-1} \frac{c_{\mathrm{IL}}}{L_{\mathrm{IL}}} \rho(t) \|\nabla \mathcal{L}_{\mathrm{IL}}(\theta_t)\| \cdot \|\nabla \mathcal{L}_{\mathrm{RL}}(\theta_t)\|$$

$$+ \frac{L_{\mathrm{RL}} c_{\mathrm{IL}}^2}{2 L_{\mathrm{IL}}^2} \cdot \frac{L_{\mathrm{IL}}(\mathcal{L}_{\mathrm{IL}}(\theta_0) - \mathcal{L}_{\mathrm{IL}}^*)}{c_{\mathrm{IL}}(1 - \frac{c_{\mathrm{IL}}}{2})} + T \frac{L_{\mathrm{RL}} c_{\mathrm{IL}}^2 \sigma_{\mathrm{IL}}^2}{2 L_{\mathrm{IL}}^2 N_{\mathrm{IL}}}$$

$$+ \bar{m} T \frac{c_{\mathrm{RL}}^2 \sigma_{\mathrm{RL}}^2}{2 L_{\mathrm{RL}} N_{\mathrm{RL}}}$$

The term with IL gradient norms can be simplified to:

$$\frac{L_{\mathrm{RL}} c_{\mathrm{IL}}^2}{2 L_{\mathrm{IL}}^2} \cdot \frac{L_{\mathrm{IL}}(\mathcal{L}_{\mathrm{IL}}(\theta_0) - \mathcal{L}_{\mathrm{IL}}^*)}{c_{\mathrm{IL}}(1 - \frac{c_{\mathrm{IL}}}{2})} = \frac{L_{\mathrm{RL}} c_{\mathrm{IL}}}{2 L_{\mathrm{IL}}} \cdot \frac{(\mathcal{L}_{\mathrm{IL}}(\theta_0) - \mathcal{L}_{\mathrm{IL}}^*)}{(1 - \frac{c_{\mathrm{IL}}}{2})}$$

By Assumption 1, $\mathcal{L}_{\mathrm{IL}}(\theta_0) - \mathcal{L}_{\mathrm{IL}}^* \leq \epsilon_{\mathrm{IL}}$, which is small. For large enough $T$, this term becomes negligible.

Define the IL regularization benefit:

$$\Delta_{\mathrm{IL-RL}} = -\sum_{t=0}^{T-1} \frac{c_{\mathrm{IL}}}{L_{\mathrm{IL}}} \rho(t) \|\nabla \mathcal{L}_{\mathrm{IL}}(\theta_t)\| \cdot \|\nabla \mathcal{L}_{\mathrm{RL}}(\theta_t)\| + \frac{c_{\mathrm{IL}}^2 \sigma_{\mathrm{IL}}^2 T}{2 L_{\mathrm{IL}} N_{\mathrm{IL}}}$$

With this, our bound becomes:

$$\bar{m} \frac{c_{\mathrm{RL}}(1 - \frac{c_{\mathrm{RL}}}{2})}{L_{\mathrm{RL}}} \frac{1}{T} \sum_{t=0}^{T-1} \|\nabla \mathcal{L}_{\mathrm{RL}}(\theta_t)\|^2 \leq \frac{\mathcal{L}_{\mathrm{RL}}(\theta_0) - \mathcal{L}_{\mathrm{RL}}^* - \Delta_{\mathrm{IL-RL}}}{T} + \bar{m} \frac{c_{\mathrm{RL}}^2 \sigma_{\mathrm{RL}}^2}{2 L_{\mathrm{RL}} N_{\mathrm{RL}}}$$

By the pigeonhole principle, there must exist at least one iteration $t^* \in \{0, 1, \ldots, T-1\}$ such that:

$$\|\nabla\mathcal{L}_{\text{RL}}(\theta_{t^*})\|^2 \leq \frac{1}{T}\sum_{t=0}^{T-1}\|\nabla\mathcal{L}_{\text{RL}}(\theta_t)\|^2$$

Therefore:

$$\min_{0 \leq t < T}\|\nabla\mathcal{L}_{\text{RL}}(\theta_t)\|^2 \leq \frac{L_{\text{RL}}(\mathcal{L}_{\text{RL}}(\theta_0) - \mathcal{L}_{\text{RL}}^* - \Delta_{\text{IL-RL}})}{c_{\text{RL}}(1 - \frac{c_{\text{RL}}}{2})\bar{m}T} + \frac{c_{\text{RL}}^2\sigma_{\text{RL}}^2}{2c_{\text{RL}}(1 - \frac{c_{\text{RL}}}{2})N_{\text{RL}}}$$

Taking expectation and adjusting the constant in the second term:

$$\min_{0 \leq t < T}\mathbb{E}[\|\nabla\mathcal{L}_{\text{RL}}(\theta_t)\|^2] \leq \frac{2(L_{\text{RL}}(\mathcal{L}_{\text{RL}}(\theta_0) - \mathcal{L}_{\text{RL}}^*) - \Delta_{\text{IL-RL}})}{c_{\text{RL}}(1 - \frac{c_{\text{RL}}}{2})\bar{m}T} + \frac{c_{\text{RL}}\sigma_{\text{RL}}^2}{(1 - \frac{c_{\text{RL}}}{2})N_{\text{RL}}}$$

For the IL performance bound, using the earlier bound on IL updates and summing over all cycles:

$$\mathcal{L}_{\text{IL}}(\theta_T) - \mathcal{L}_{\text{IL}}(\theta_0) \leq -\sum_{t=0}^{T-1}\frac{c_{\text{IL}}(1 - \frac{c_{\text{IL}}}{2})}{L_{\text{IL}}}\|\nabla\mathcal{L}_{\text{IL}}(\theta_t)\|^2 + \frac{c_{\text{IL}}^2\sigma_{\text{IL}}^2 T}{2L_{\text{IL}}N_{\text{IL}}}$$

Combining with Assumption 1:

$$\mathcal{L}_{\text{IL}}(\theta_T) - \mathcal{L}_{\text{IL}}(\theta^*) = \mathcal{L}_{\text{IL}}(\theta_T) - \mathcal{L}_{\text{IL}}(\theta_0) + \mathcal{L}_{\text{IL}}(\theta_0) - \mathcal{L}_{\text{IL}}(\theta^*)$$

$$\leq -\sum_{t=0}^{T-1}\frac{c_{\text{IL}}(1 - \frac{c_{\text{IL}}}{2})}{L_{\text{IL}}}\|\nabla\mathcal{L}_{\text{IL}}(\theta_t)\|^2 + \frac{c_{\text{IL}}^2\sigma_{\text{IL}}^2 T}{2L_{\text{IL}}N_{\text{IL}}} + \epsilon_{\text{IL}}$$

Additionally, by the $L_{\text{IL}}$-smoothness of the IL objective:

$$\mathcal{L}_{\text{IL}}(\theta_T) - \mathcal{L}_{\text{IL}}(\theta_0) \leq \frac{L_{\text{IL}}}{2}\|\theta_T - \theta_0\|^2$$

Combining these bounds:

$$\mathcal{L}_{\text{IL}}(\theta_T) - \mathcal{L}_{\text{IL}}(\theta^*) \leq \epsilon_{\text{IL}} + \frac{L_{\text{IL}}}{2}\|\theta_T - \theta_0\|^2 - \sum_{t=0}^{T-1}\frac{c_{\text{IL}}(1 - \frac{c_{\text{IL}}}{2})}{L_{\text{IL}}}\|\nabla\mathcal{L}_{\text{IL}}(\theta_t)\|^2$$

This shows that the periodic IL updates in interleaved training help maintain good IL performance compared to RL-only training. □

## E    PROOF OF THEOREM 1

*Proof.* To find the optimal ratio $m(t)$ at iteration $t$, we want to maximize the progress per update. From our analysis in Theorem 2, the progress for one complete cycle is:

$$\mathcal{L}_{\text{RL}}(\theta_t) - \mathcal{L}_{\text{RL}}(\theta_{t+1}) \approx m(t)\frac{c_{\text{RL}}(1 - \frac{c_{\text{RL}}}{2})}{L_{\text{RL}}}\|\nabla\mathcal{L}_{\text{RL}}(\theta_t)\|^2$$

$$- \frac{c_{\text{IL}}}{L_{\text{IL}}}\rho(t)\|\nabla\mathcal{L}_{\text{IL}}(\theta_t)\| \cdot \|\nabla\mathcal{L}_{\text{RL}}(\theta_t)\|$$

$$- \frac{L_{\text{RL}}c_{\text{IL}}^2\sigma_{\text{IL}}^2}{2L_{\text{IL}}^2 N_{\text{IL}}} - m(t)\frac{c_{\text{RL}}^2\sigma_{\text{RL}}^2}{2L_{\text{RL}}N_{\text{RL}}}$$

Since each cycle consists of $1 + m(t)$ updates, the progress per update is:

$$\frac{\mathcal{L}_{\text{RL}}(\theta_t) - \mathcal{L}_{\text{RL}}(\theta_{t+1})}{1 + m(t)}$$

To find the optimal $m(t)$, we differentiate this expression with respect to $m(t)$ and set it to zero. Let's denote:

$$A = \frac{c_{\text{RL}}(1 - \frac{c_{\text{RL}}}{2})}{L_{\text{RL}}}\|\nabla\mathcal{L}_{\text{RL}}(\theta_t)\|^2$$

$$B = \frac{c_{\text{IL}}}{L_{\text{IL}}}\rho(t)\|\nabla\mathcal{L}_{\text{IL}}(\theta_t)\| \cdot \|\nabla\mathcal{L}_{\text{RL}}(\theta_t)\| + \frac{L_{\text{RL}}c_{\text{IL}}^2\sigma_{\text{IL}}^2}{2L_{\text{IL}}^2 N_{\text{IL}}}$$

$$C = \frac{c_{\text{RL}}^2\sigma_{\text{RL}}^2}{2L_{\text{RL}}N_{\text{RL}}}$$

Then the progress per update is:

$$\frac{mA - B - mC}{1 + m}$$

Differentiating with respect to $m$:

$$\frac{d}{dm}\left(\frac{mA - B - mC}{1 + m}\right) = \frac{(A - C)(1 + m) - (mA - B - mC)}{(1 + m)^2}$$

$$= \frac{A - C + mA - mC - mA + B + mC}{(1 + m)^2}$$

$$= \frac{A - C + B}{(1 + m)^2}$$

For this to be zero, we need $A - C + B = 0$, which is not possible in general if $A > C$ (which is the case when the RL objective has room for improvement). Therefore, the derivative is always positive or always negative.

Since we're looking for a maximum, we need to check the second derivative:

$$\frac{d^2}{dm^2}\left(\frac{mA - B - mC}{1 + m}\right) = \frac{d}{dm}\left(\frac{A - C + B}{(1 + m)^2}\right)$$

$$= (A - C + B) \cdot \frac{d}{dm}\left(\frac{1}{(1 + m)^2}\right)$$

$$= (A - C + B) \cdot \left(-\frac{2}{(1 + m)^3}\right)$$

$$= -\frac{2(A - C + B)}{(1 + m)^3}$$

When $A - C > B$, the second derivative is negative, indicating a maximum. In this case, the progress per update increases with $m$, and the optimal $m(t)$ would be as large as possible.

However, for practical reasons, we want to maintain some IL updates, so we need to find a suitable $m(t)$ that balances progress and regularization. One approach is to equate the progress from RL updates with the potential negative impact of the IL update:

$$m(t)\frac{c_{\text{RL}}(1 - \frac{c_{\text{RL}}}{2})}{L_{\text{RL}}}\|\nabla\mathcal{L}_{\text{RL}}(\theta_t)\|^2 \approx \frac{c_{\text{IL}}}{L_{\text{IL}}}\rho(t)\|\nabla\mathcal{L}_{\text{IL}}(\theta_t)\| \cdot \|\nabla\mathcal{L}_{\text{RL}}(\theta_t)\| + \frac{L_{\text{RL}}c_{\text{IL}}^2\sigma_{\text{IL}}^2}{2L_{\text{IL}}^2 N_{\text{IL}}}$$

Solving for $m(t)$:

$$m(t) \approx \frac{\frac{c_{\text{IL}}}{L_{\text{IL}}}\rho(t)\|\nabla\mathcal{L}_{\text{IL}}(\theta_t)\| \cdot \|\nabla\mathcal{L}_{\text{RL}}(\theta_t)\| + \frac{L_{\text{RL}}c_{\text{IL}}^2\sigma_{\text{IL}}^2}{2L_{\text{IL}}^2 N_{\text{IL}}}}{\frac{c_{\text{RL}}(1 - \frac{c_{\text{RL}}}{2})}{L_{\text{RL}}}\|\nabla\mathcal{L}_{\text{RL}}(\theta_t)\|^2}$$

$$= \frac{L_{\text{RL}}c_{\text{IL}}\rho(t)\|\nabla\mathcal{L}_{\text{IL}}(\theta_t)\|}{L_{\text{IL}}c_{\text{RL}}(1 - \frac{c_{\text{RL}}}{2})\|\nabla\mathcal{L}_{\text{RL}}(\theta_t)\|} + \frac{L_{\text{RL}}^2 c_{\text{IL}}^2\sigma_{\text{IL}}^2}{2L_{\text{IL}}^2 N_{\text{IL}}c_{\text{RL}}(1 - \frac{c_{\text{RL}}}{2})\|\nabla\mathcal{L}_{\text{RL}}(\theta_t)\|^2}$$

When gradients are opposing ($\rho(t) > 0$), this can give a reasonably large $m(t)$. When gradients are align*ed ($\rho(t) < 0$), the optimal $m(t)$ would be smaller.

A more practical approach is to use a square root formula that balances these factors:

$$m_{\text{opt}}(t) = \max \left\{ 1, \sqrt{\frac{\|\nabla\mathcal{L}_{\text{RL}}(\theta_t)\|^2}{\rho(t)\|\nabla\mathcal{L}_{\text{IL}}(\theta_t)\| \cdot \|\nabla\mathcal{L}_{\text{RL}}(\theta_t)\| - \frac{c_{\text{IL}}L_{\text{RL}}\sigma_{\text{IL}}^2}{2L_{\text{IL}}^2 N_{\text{IL}}}}} \right\}$$

This formula ensures that: 1. $m(t)$ is at least 1 (we always do at least one RL update per IL update) 2. $m(t)$ increases when RL gradients are large relative to IL gradients 3. $m(t)$ increases when gradients oppose each other ($\rho(t) > 0$ and large) 4. $m(t)$ decreases when gradients align* ($\rho(t) < 0$)

The specific constants may need to be adjusted based on empirical observations, but this formula provides a theoretically justified starting point for adaptive interleaving. $\qquad\square$

## F    PROOF OF THEOREM 2

*Proof.* From Theorem 1, the number of iterations required for RL-only training to reach a target accuracy $\min_{0 \le t < T} \|\nabla\mathcal{L}_{\text{RL}}(\theta_t)\|^2 \le \epsilon$ is:

$$T_{\text{RL-only}} \approx \frac{2L_{\text{RL}}(\mathcal{L}_{\text{RL}}(\theta_0) - \mathcal{L}_{\text{RL}}^*)}{c_{\text{RL}}(1 - \frac{c_{\text{RL}}}{2})\epsilon}$$

From Theorem 2, the number of cycles required for interleaved $1{:}m(t)$ training to reach the same accuracy is:

$$T_{\text{interleaved, cycles}} \approx \frac{2(L_{\text{RL}}(\mathcal{L}_{\text{RL}}(\theta_0) - \mathcal{L}_{\text{RL}}^*) - \Delta_{\text{IL}-\text{RL}})}{c_{\text{RL}}(1 - \frac{c_{\text{RL}}}{2})\bar{m}\epsilon}$$

Since each cycle consists of $1 + m(t)$ updates, the total number of updates required for interleaved training is:

$$T_{\text{interleaved, updates}} \approx (1 + \bar{m})T_{\text{interleaved, cycles}}$$
$$\approx (1 + \bar{m})\frac{2(L_{\text{RL}}(\mathcal{L}_{\text{RL}}(\theta_0) - \mathcal{L}_{\text{RL}}^*) - \Delta_{\text{IL}-\text{RL}})}{c_{\text{RL}}(1 - \frac{c_{\text{RL}}}{2})\bar{m}\epsilon}$$

For a fair comparison, we compare the total number of updates required by both methods. The ratio is:

$$\frac{T_{\text{RL-only}}}{T_{\text{interleaved, updates}}} = \frac{\frac{2L_{\text{RL}}(\mathcal{L}_{\text{RL}}(\theta_0) - \mathcal{L}_{\text{RL}}^*)}{c_{\text{RL}}(1 - \frac{c_{\text{RL}}}{2})\epsilon}}{(1 + \bar{m})\frac{2(L_{\text{RL}}(\mathcal{L}_{\text{RL}}(\theta_0) - \mathcal{L}_{\text{RL}}^*) - \Delta_{\text{IL}-\text{RL}})}{c_{\text{RL}}(1 - \frac{c_{\text{RL}}}{2})\bar{m}\epsilon}}$$
$$= \frac{\bar{m}}{1 + \bar{m}} \cdot \frac{L_{\text{RL}}(\mathcal{L}_{\text{RL}}(\theta_0) - \mathcal{L}_{\text{RL}}^*)}{L_{\text{RL}}(\mathcal{L}_{\text{RL}}(\theta_0) - \mathcal{L}_{\text{RL}}^*) - \Delta_{\text{IL}-\text{RL}}}$$

When $\Delta_{\text{IL}-\text{RL}} > 0$ (positive regularization benefit) and $\bar{m} > 1$, this ratio can be greater than 1, indicating that interleaved training requires fewer total updates than RL-only training to achieve the same level of accuracy.

Specifically, if we define the relative regularization benefit:

$$\beta = \frac{\Delta_{\text{IL}-\text{RL}}}{L_{\text{RL}}(\mathcal{L}_{\text{RL}}(\theta_0) - \mathcal{L}_{\text{RL}}^*)}$$

Then the ratio becomes:

$$\frac{T_{\text{RL-only}}}{T_{\text{interleaved, updates}}} = \frac{\bar{m}}{1 + \bar{m}} \cdot \frac{1}{1 - \beta}$$

For interleaved training to be more efficient than RL-only training, we need:

$$\frac{\bar{m}}{1+\bar{m}} \cdot \frac{1}{1-\beta} > 1$$

This is satisfied when:

$$\beta > 1 - \frac{\bar{m}}{1+\bar{m}} = \frac{1}{1+\bar{m}}$$

For example, with $\bar{m} = 3$, interleaved training is more efficient when $\beta > \frac{1}{4}$, i.e., when the regularization benefit is at least 25% of the potential RL improvement. □

## F.1 INTERPRETING THE EFFICIENCY ADVANTAGE

Our theoretical analysis requires careful interpretation to properly understand the efficiency relationship between IN–RIL and RL-only methods. In what follows, we further examine the key results and their implications.

### F.1.1 EFFICIENCY RATIO

From our theoretical analysis, we derived the efficiency ratio comparing RL-only updates to total interleaved updates:

$$\frac{T_{\text{RL-only}}}{T_{\text{IN–RIL,total}}} = \frac{m_{\text{opt}}}{1+m_{\text{opt}}} \cdot \frac{L_{\text{RL}}(\mathcal{L}_{\text{RL}}(\theta_0) - \mathcal{L}_{\text{RL}}^*)}{L_{\text{RL}}(\mathcal{L}_{\text{RL}}(\theta_0) - \mathcal{L}_{\text{RL}}^*) - \Delta_{\text{IL−RL}}}$$

Let's examine this ratio's behavior in different scenarios:

1. **As $m_{\text{opt}} \to \infty$**: The term $\frac{m_{\text{opt}}}{1+m_{\text{opt}}} \to 1$, and the ratio approaches $\frac{L_{\text{RL}}(\mathcal{L}_{\text{RL}}(\theta_0) - \mathcal{L}_{\text{RL}}^*)}{L_{\text{RL}}(\mathcal{L}_{\text{RL}}(\theta_0) - \mathcal{L}_{\text{RL}}^*) - \Delta_{\text{IL−RL}}}$

2. **When $\Delta_{\text{IL−RL}} = 0$**: The ratio simplifies to $\frac{m_{\text{opt}}}{1+m_{\text{opt}}}$, which is always less than 1, indicating that IN–RIL requires more updates

3. **When $\Delta_{\text{IL−RL}} > 0$**: The ratio may exceed 1 if the regularization benefit is sufficiently large

To properly assess when IN–RIL is more efficient (ratio > 1), we need to solve:

$$\frac{m_{\text{opt}}}{1+m_{\text{opt}}} \cdot \frac{L_{\text{RL}}(\mathcal{L}_{\text{RL}}(\theta_0) - \mathcal{L}_{\text{RL}}^*)}{L_{\text{RL}}(\mathcal{L}_{\text{RL}}(\theta_0) - \mathcal{L}_{\text{RL}}^*) - \Delta_{\text{IL−RL}}} > 1$$

Rearranging, we get:

$$\Delta_{\text{IL−RL}} > L_{\text{RL}}(\mathcal{L}_{\text{RL}}(\theta_0) - \mathcal{L}_{\text{RL}}^*) \cdot \left(1 - \frac{m_{\text{opt}}}{1+m_{\text{opt}}}\right) = \frac{L_{\text{RL}}(\mathcal{L}_{\text{RL}}(\theta_0) - \mathcal{L}_{\text{RL}}^*)}{1+m_{\text{opt}}}$$

### F.1.2 KEY INSIGHTS

1. **Asymptotic Behavior**: As $m_{\text{opt}} \to \infty$, the efficiency condition approaches $\Delta_{\text{IL−RL}} > 0$. This means with very large interleaving ratios, even a small positive regularization benefit makes IN–RIL more efficient.

2. **Impact of Interleaving Ratio**: For any finite $m_{\text{opt}}$, IN–RIL includes an overhead factor of $\frac{1+m_{\text{opt}}}{m_{\text{opt}}}$ that must be overcome by the regularization benefit.

3. **Alternative View**: We can rewrite the ratio as:
$$\frac{T_{\text{RL-only}}}{T_{\text{IN–RIL,total}}} = \frac{L_{\text{RL}}(\mathcal{L}_{\text{RL}}(\theta_0) - \mathcal{L}_{\text{RL}}^*)}{L_{\text{RL}}(\mathcal{L}_{\text{RL}}(\theta_0) - \mathcal{L}_{\text{RL}}^*) - \Delta_{\text{IL−RL}} + \frac{L_{\text{RL}}(\mathcal{L}_{\text{RL}}(\theta_0) - \mathcal{L}_{\text{RL}}^*)}{m_{\text{opt}}}}$$

This form explicitly shows the penalty term $\frac{L_{\text{RL}}(\mathcal{L}_{\text{RL}}(\theta_0) - \mathcal{L}_{\text{RL}}^*)}{m_{\text{opt}}}$, which decreases as $m_{\text{opt}}$ increases.

### F.1.3 PRACTICAL IMPLICATIONS

Our theoretical analysis provides important practical guidance:

1. **Optimal Interleaving Ratio**: There is a trade-off in setting $m_{\text{opt}}$:

   - Small $m_{\text{opt}}$ (e.g., $m_{\text{opt}} = 1$): IN–RIL needs $\Delta_{\text{IL}-\text{RL}} > \frac{L_{\text{RL}}(\mathcal{L}_{\text{RL}}(\theta_0) - \mathcal{L}^*_{\text{RL}})}{2}$ to be more efficient

   - Large $m_{\text{opt}}$ (e.g., $m_{\text{opt}} = 9$): IN–RIL needs $\Delta_{\text{IL}-\text{RL}} > \frac{L_{\text{RL}}(\mathcal{L}_{\text{RL}}(\theta_0) - \mathcal{L}^*_{\text{RL}})}{10}$ to be more efficient

   - Very large $m_{\text{opt}}$: IN–RIL approaches the behavior of RL-only but retains modest regularization benefits

2. **Environment Interaction Efficiency**: If we consider only RL updates (environment interactions):

$$\frac{T_{\text{RL-only}}}{T_{\text{IN–RIL,RL}}} = \frac{L_{\text{RL}}(\mathcal{L}_{\text{RL}}(\theta_0) - \mathcal{L}^*_{\text{RL}})}{L_{\text{RL}}(\mathcal{L}_{\text{RL}}(\theta_0) - \mathcal{L}^*_{\text{RL}}) - \Delta_{\text{IL}-\text{RL}}}$$

   This ratio is greater than 1 whenever $\Delta_{\text{IL}-\text{RL}} > 0$, showing that IN–RIL always requires fewer environment interactions when there is any positive regularization benefit.

3. **Practical Recommendation**: Based on our empirical evaluations across multiple benchmarks, interleaving ratios between 3 and 5 typically provide the best balance. This align*s with our theory: with $m_{\text{opt}} = 4$, IN–RIL is more computationally efficient when $\Delta_{\text{IL}-\text{RL}} > \frac{L_{\text{RL}}(\mathcal{L}_{\text{RL}}(\theta_0) - \mathcal{L}^*_{\text{RL}})}{5}$, a threshold often satisfied in practice.

### F.1.4 EMPIRICAL VALIDATION

Our experiments confirm the theoretical predictions:

- Across our benchmark tasks, IN–RIL demonstrated significant improvements in sample efficiency, significantly reducing required interactions

- The largest efficiency gains occurred in tasks where the estimated regularization benefit $\Delta_{\text{IL}-\text{RL}}$ was highest, exactly as predicted by our theory

- The relationship between efficiency gains and interleaving ratio matched our theoretical expectations, with diminishing returns for very large ratios

## G EXTENDED RELATED WORKS

Imitation learning (IL) Brohan et al. (2022); Kim et al. (2024); Chi et al. (2023); Fu et al. (2024); Lee et al. (2024) and reinforcement learning (RL) Kalashnikov et al. (2018); Han et al. (2023); Hafner et al. (2023); Ren et al. (2024); Wu et al. (2023); Ankile et al. (2024) have been widely studied in robotics. IL assumes access to expert demonstrations and is generally more stable to train Chi et al. (2023); Shafiullah et al. (2022), but it suffers from distribution shifts and often fails to generalize beyond demonstrations Rajeswaran et al. (2018). In addition, collecting high-quality expert data can be labor-intensive and costly, sometimes requiring hundreds of demonstrations per task Zhao et al. (2024) through teleoperation Fu et al. (2024), or VR equipments Chuang et al. (2024). RL enables agents to explore and self-improve, potentially overcoming IL limitations of labor-intensive data collection and generalization. However, RL is notoriously sample-inefficient Song et al. (2022), especially for long-horizon tasks with sparse rewards Gupta et al. (2019), where agents may easily fail to learn via random exploration. Recent works propose combining IL and RL in a two-stage pipeline: IL is first used to pre-train a reasonable policy, followed by RL fine-tuning to further improve generalization via exploration Ren et al. (2024); Ankile et al. (2024); Hu et al. (2023). The same paradigm was also applied to LLM fine-tuning Guo et al. (2025). IN–RIL moves beyond the two-stage paradigm, and shows that the data used for pre-training, even after pre-training plateaus, is still valuable in improving sample-efficiency and stability during RL fine-tuning.

# H SUPPLEMENTARY EXPERIMENTS

## H.1 ROBOMIMIC AND GYM

### H.1.1 ADDITIONAL RESULTS

We have reported success rates of Robomimic tasks in Table 1 Additionally, Table 3 reports the rewards metrics for Robomimic tasks.

Table 3: Rewards on Robomimic and Gym tasks. **Bold** values indicate the best within the DPPO group or the IDQL group. *Italic* values indicate the overall best across all methods.

| Task | IN-RIL (DPPO) | DPPO | IN-RIL (IDQL) | IDQL | BC Reg | DIPO | AWR |
|------|---------------|------|---------------|------|--------|------|-----|
| Transport | ***323.42*** | 299.98 | **267.27** | 12.53 | 101.45 | 31.91 | 31.91 |
| Can | 204.60 | ***207.10*** | **193.88** | 184.64 | 152.12 | 110.88 | 78.06 |
| Lift | 136.80 | **139.75** | **205.00** | 162.49 | 52.77 | 64.56 | *205.81* |
| Square | **237.70** | 233.80 | ***245.70*** | 158.34 | 116.43 | 106.73 | 107.84 |
| Walker2D | **4139** | 3786 | 4186 | **4248** | 3523 | 3715 | *4250* |
| Hopper | **2930** | 2929 | ***3042*** | 2988 | 2896 | 2938 | 1427 |
| HalfCheetah | 4887 | ***5011*** | **4742** | 4671 | 4532 | 4644 | 4611 |

For each task and fine-tuning method in Table 1, we have reported the best success rates or rewards across all checkpoints. Since some methods collapsed after performance peaked (e.g., DPPO on `Hopper`, we include Section H.1.1 to show the last performance metric at the timestep budget. IN–RIL consistently outperforms on most tasks, and IN–RIL is overall stable throughout training.

| Task | IN-RIL (DPPO) | DPPO | IN-RIL (IDQL) | IDQL | BC Reg | DIPO | AWR |
|------|---------------|------|---------------|------|--------|------|-----|
| Transport | ***0.91*** | 0.78 | **0.85** | 0.00 | 0.37 | 0.00 | 0.03 |
| Can | **0.99** | 0.99 | 0.96 | ***1.00*** | 0.92 | 0.94 | 0.34 |
| Lift | 0.93 | ***1.00*** | 0.98 | **0.99** | 0.97 | 0.96 | 0.94 |
| Square | **0.91** | 0.86 | ***0.96*** | 0.78 | 0.58 | 0.59 | 0.38 |
| Walker2D | **4044** | 3746 | 4151 | ***4245*** | 3239 | 3046 | 4232 |
| Hopper | **2890** | 2517 | ***3024*** | 2907 | 2664 | 2753 | 1381 |
| HalfCheetah | 4820 | ***4854*** | **4715** | 4602 | 4393 | 4644 | 4611 |

Table 4: Last performance metrics at budget for all methods on Robomimic (success rates) and Gym (rewards) tasks. **Bold** values indicate the best within its group. *Italic* values indicate the overall best across all methods.

### H.1.2 SELECTION OF $m$

It has been shown in Figure 6 that, $m$ impacts different tasks in a different way, while IN–RIL is overall robust to the selection $m$. For example, our ablation shows that $2 \leq m \leq 30$ for IDQL on `Transport` all yield superior performance. Here we report one of the optimal $m$ in Section H.1.2.

| Task | $m$ (IN–RIL + DPPO) | $m$ (IN–RIL + IDQL) |
|------|---------------------|---------------------|
| Transport | 10 | 10 |
| Can | 5 | 10 |
| Lift | 10 | 7 |
| Square | 30 | 5 |
| Walker2D | 30 | 10 |
| Hopper | 10 | 10 |
| HalfCheetah | 10 | 10 |

Table 5: Selection of $m$ for IN–RIL in experiments.

## H.2 FurnitureBench

### H.2.1 Additional Results

We show the pre-training results for different policy parameterizations in Table 6. Gaussian policies without action chunking are unable to solve FurnitureBench tasks. Gaussian without action chunking enables the agent to learn reasonable policies on most tasks, while DP still yields the strongest performance.

| | Policy Parameterization | OneLeg Low | OneLeg Med | Lamp Low | Lamp Med | RoundTable | MugRack | PegInHole |
|---|---|---|---|---|---|---|---|---|
| | Gaussian w/ Action Chunking | 0.38 | 0.17 | **0.07** | 0.02 | 0.01 | 0.14 | 0.02 |
| BC | Gaussian w/o Action Chunking | 0.0 | 0.0 | 0.0 | 0.0 | 0.0 | 0.0 | 0.0 |
| | DP | **0.47** | **0.28** | 0.05 | **0.1** | **0.10** | **0.19** | **0.03** |

Table 6: Success rates across FurnitureBench tasks Ankile et al. (2024); Heo et al. (2023) using pre-trained policies.

We report two extra tasks, Mug-Rack and Peg-in-Hole provided by ResiP. IN–RIL matches the ultimate performance, and exceeds in sample efficiency on Peg-in-Hole that was shown in the reward curves in Figure 5.

| Task | IN-RIL (Residual PPO) | Residual PPO | IDQL |
|------|----------------------|--------------|------|
| Peg-in-Hole | 0.93 | 0.92 | 0.01 |
| Mug-Rack | 0.85 | 0.85 | 0.16 |

Table 7: Comparing IN–RIL with other RL fine-tuning algorithms on FurnitureBench. Bold values indicate the best of all.

### H.2.2 Selection of $m$

| Task | $m$ |
|------|-----|
| One-Leg (Low) | 5 |
| One-Leg (Med) | 5 |
| Lamp (Low) | 4 |
| Lamp (Med) | 10 |
| Round-Table (Low) | 5 |
| Mug-Rack | 5 |
| Peg-in-Hole | 5 |

Table 8: Selection of $m$ for IN–RIL in FurnitureBench experiments using residual PPO.

### H.2.3 IN–RIL FROM SCRATCH

Beyond using IN–RIL for fine-tuning, we also consider using IN–RIL to learn policies from scratch. We compare IN–RIL trained from scratch against RL fine-tuning following a pre-trained policy. We notice that IN–RIL policies learned from scratch catch up with RL fine-tuning quickly, and learns stably throughout the process. Similar to what we observed in IN–RIL for fine-tuning, IN–RIL behaves more stable than residual PPO at around $4 \times 10^8$ steps, where residual PPO may slightly degrade.

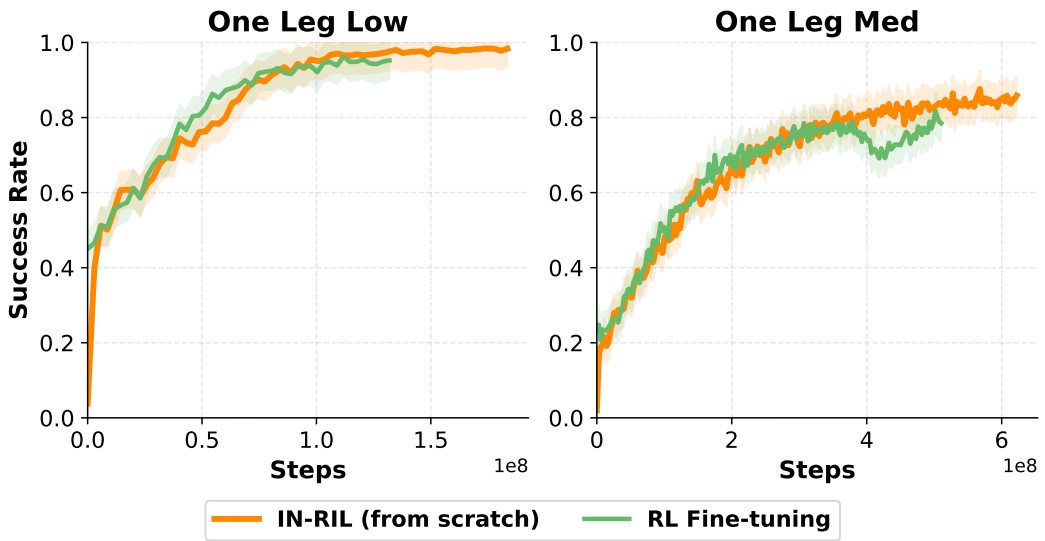

Figure 8: Comparing IN–RIL trained from scratch (orange curves that start from $0\%$ success rates) to RL fine-tuning (green curves that starts from pre-trained policy success rates).

### H.2.4 TASK ROLLOUTS

In this section, we show examples of policy rollouts, including successful and failed ones, of IN–RIL agents on all the FurnitureBench tasks. The tasks are challenging for RL agents because of their long-horizon and sparse-reward natures. The agent needs to assemble different parts, and and assembly requires high precision.

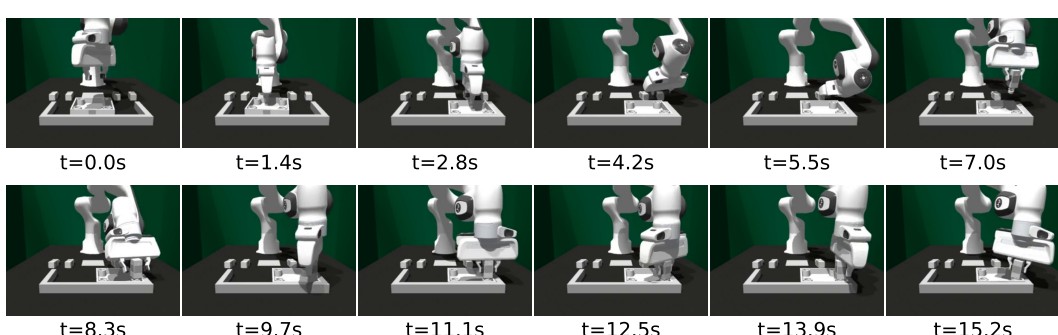

Figure 9: A successful rollout example of the `One-Leg` (Low) furniture assembly task

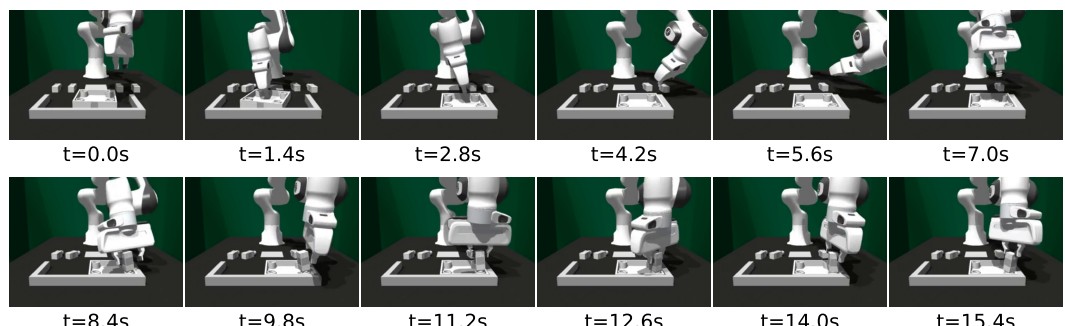

Figure 10: A successful rollout example of the `One-Leg` (Med) furniture assembly task

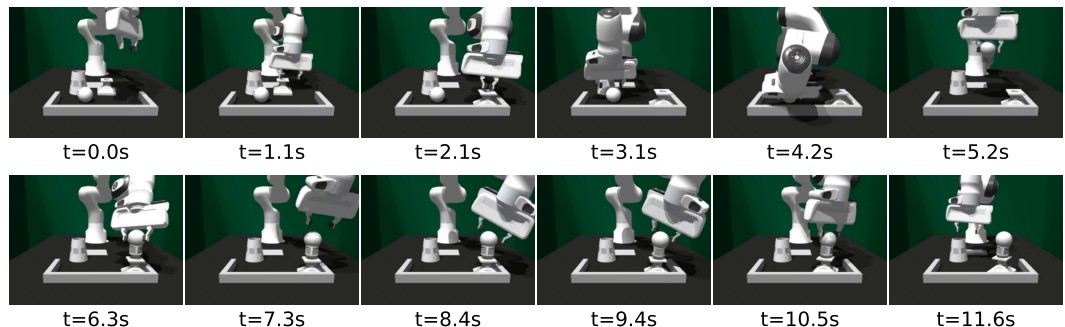

Figure 11: A successful rollout example of the `Lamp` (Low) assembly task

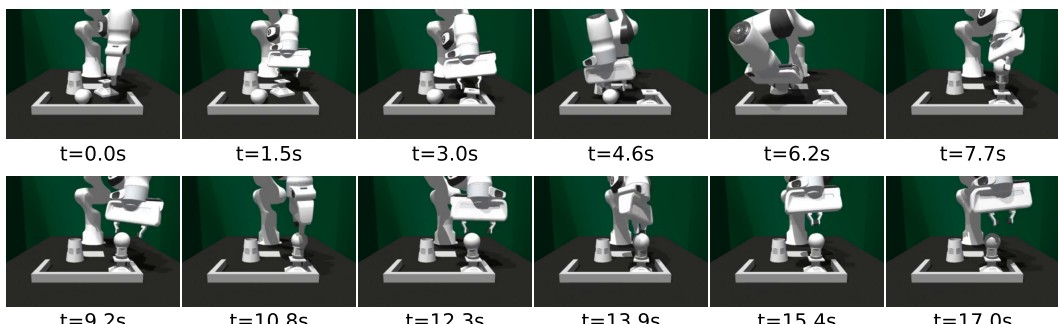

Figure 12: A successful rollout example of the `Lamp` (Med) task

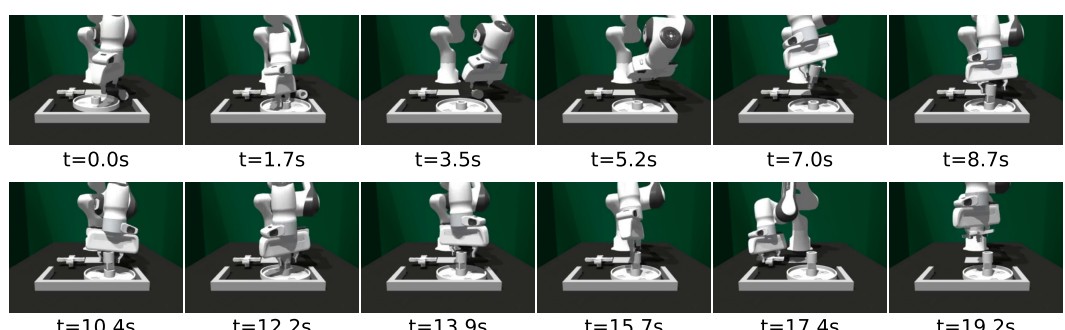

Figure 13: A successful rollout example of the `Round-Table` assembly task

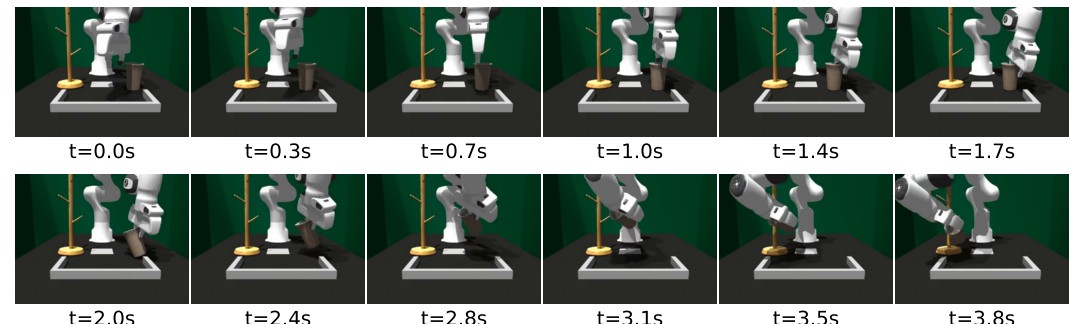

Figure 14: A successful rollout example of the `Mug-Rack` task

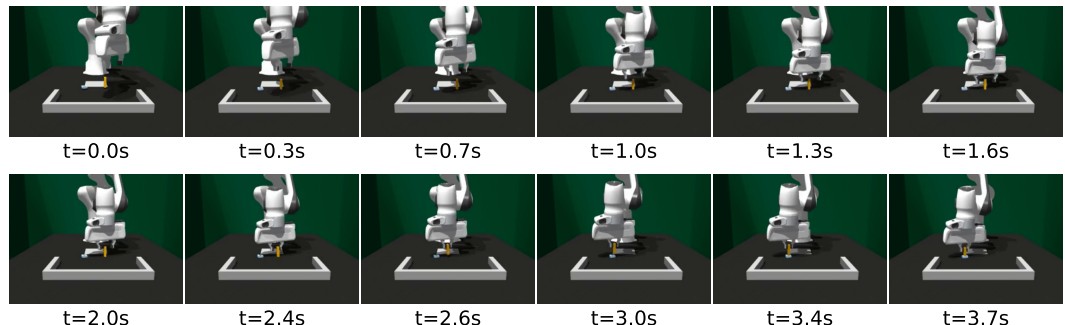

Figure 15: A successful rollout example of the `Peg-in-Hole` task

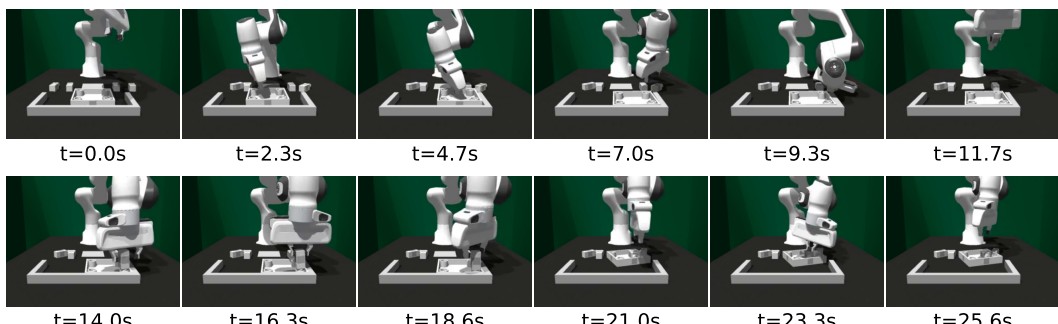

Figure 16: A failed rollout example of the `One-Leg` (Med) furniture assembly task

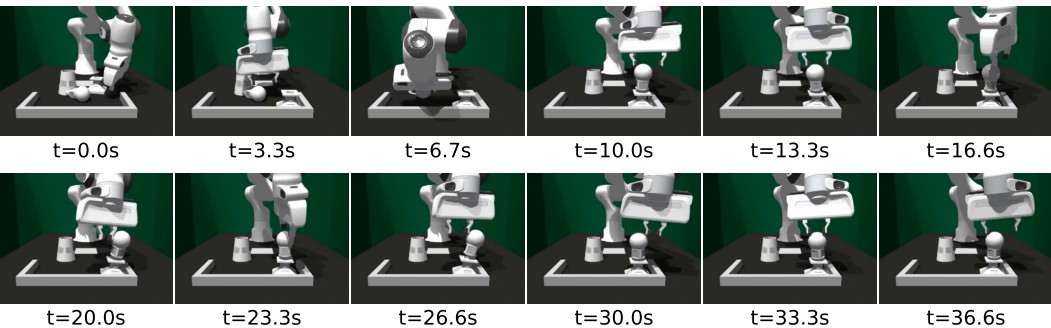

Figure 17: A failed rollout example of the `Lamp` (Med) assembly task

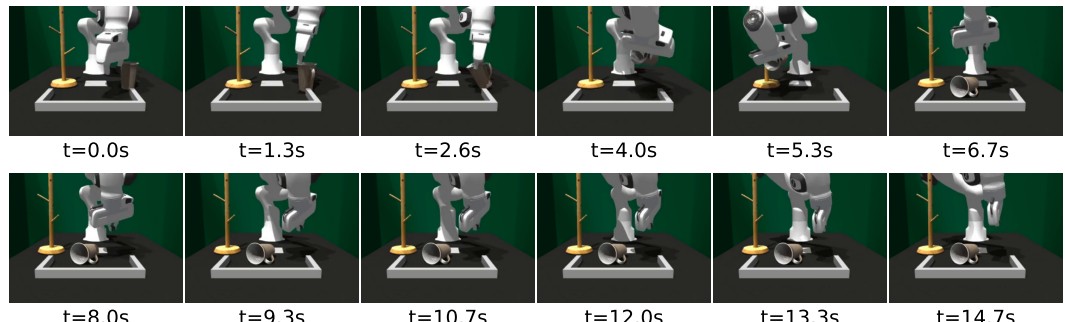

Figure 18: A failed rollout example of the `Mug-Rack` assembly task

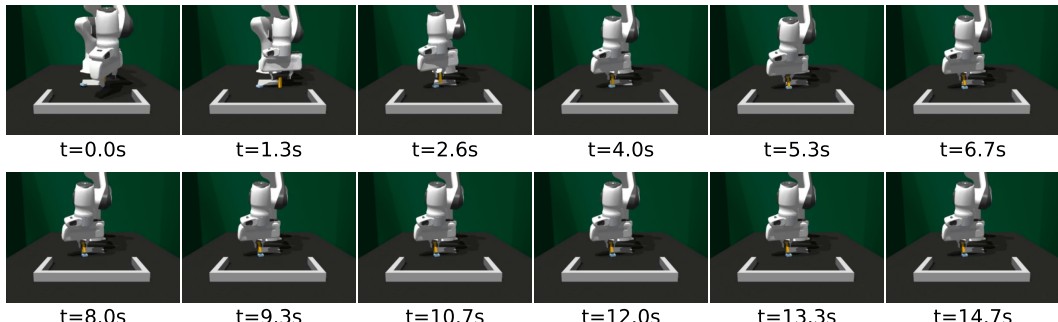

Figure 19: A failed rollout example of the `Peg-in-Hole` task

## H.3 OVERVIEW OF TASKS

We evaluate our method across a diverse set of continuous control benchmarks that span locomotion and manipulation. Our experiments include 14 tasks drawn from three widely-used suites: Robomimic, FurnitureBench, and OpenAI Gym. Each task presents a unique challenge—ranging from dense-reward locomotion to long-horizon assembly under sparse rewards. Table 9 summarizes the observation/action dimensionality, horizon, and reward sparsity for each task.

| Benchmark | Task | Obs Dim | Act Dim | Max Episode Len | Sparse Rewards |
|-----------|------|---------|---------|-----------------|----------------|
| Robomimic | Lift | 19 | 7 | 300 | Yes |
| | Can | 23 | 7 | 300 | Yes |
| | Square | 23 | 7 | 400 | Yes |
| | Transport | 59 | 14 | 800 | Yes |
| Gym | Hopper-v2 | 11 | 3 | 1000 | No |
| | Walker2D-v2 | 17 | 6 | 1000 | No |
| | HalfCheetah-v2 | 17 | 6 | 1000 | No |
| FurnitureBench | One-Leg | 58 | 10 | 700 | Yes |
| | Lamp | 44 | 10 | 1000 | Yes |
| | Round-Table | 44 | 10 | 1000 | Yes |
| | Mug-Rack | 44 | 10 | 400 | Yes |
| | Peg-in-Hole | 44 | 10 | 400 | Yes |

Table 9: Task specifications including observation/action dimensions, time horizon, and reward sparsity. All tasks use state-based input only.

We use FurnitureBench simulation implementation released by ResiP Ankile et al. (2024). `Mug-Rack` and `Peg-in-Hole` are two new tasks created by ResiP. The FurnitureBench tasks are

among the most challenging in our evaluation, featuring long-horizon multi-stage assembly with sparse binary rewards. Each task involves interacting with multiple parts, precise insertions, and coordinated screw actions using a 7-DoF Franka Emika Panda arm operating at 10Hz. Table 10 details the task-specific attributes. We follow same settings as the ResiP paper. All tasks are designed with sparse, binary stage-completion rewards, where the agent receives a reward of 1 upon completing specific assembly stages.

| Attribute | One-Leg | Round-Table | Lamp | Mug-Rack | Peg-in-Hole |
|---|---|---|---|---|---|
| # Rewards | 1 | 2 | 2 | 1 | 1 |
| # Parts to Assemble | 2 | 3 | 3 | 2 | 2 |
| Precise Insertions | 1 | 2 | 1 | 0 | 1 |
| Screwing Steps | 1 | 2 | 1 | 0 | 0 |
| Occluded Insertion | no | yes | no | no | yes |
| Precise Grasping | no | yes | no | no | yes |

Table 10: FurnitureBench task attributes.

# I    MODELLING & TRAINING

We build IN–RIL and all the baselines upon two open-source codebases, the official DPPO implementation[1] Ren et al. (2024), and residual PPO implementation for precise assembly[2] Ankile et al. (2024).

We use a consistent time-step budget across all fine-tuning methods on each task, and train the models on NVIDIA RTX 4090 GPUs. FurnitureBench can be highly parallelized — 1 GPU-hour yields roughly 2e7 environment steps with 1,024 parallel environments. Since it is sparse-reward and particularly challenging, it requires significantly more environment steps than Robomimic or Gym. Our training typically took 15 to 40 GPU hours as our fine-tuning time step budgets for furniture tasks range from 3e8 to 8e8 steps. Gym tasks require about 3 GPU-hours per 1e7 steps with 40 environments. Since Gym tasks are easier with dense rewards, we set task budgets to 2e7–3e7 steps, and each model training can take up to 10 GPU hours. Robomimic tasks take up to 6 GPU-hours for every 1e7 steps with 50 parallel environments. Since we conduct periodic batch-wise gradient surgery for IN–RIL when integrated with DPPO or IDQL, IL updates introduce additional overheads to training time. For example, training DPPO using $2e7$ steps takes around 5.5 GPU hours, and IN–RIL using $m = 10$ takes 6.5 GPU hour.

## I.1    RESIDUAL PPO

Residual PPO augments a chunked IL policy with a lightweight, timestep-level corrective Gaussian policy trained via PPO. At each control step $i$ within a base action chunk of length $T_a$, the residual policy observes the concatenated state and base action,

$$s_{t+i}^{\text{res}} = \left[ s_{t+i},\, a_{t+i}^{\text{base}} \right],$$

and outputs a corrective action $a_{t+i}^{\text{res}}$, yielding the final action

$$a_{t+i} = a_{t+i}^{\text{base}} + a_{t+i}^{\text{res}}.$$

### I.1.1    HYPER-PARAMETERS

We follow the same training hyper-parameters (shown in Table 11) and networks (shown in Table 12) as our baseline residual PPO Ankile et al. (2024). For online fine-tuning, we keep using the same RL hyper-parameters as RL-only fine-tuning; and using fixed batch size and learning rates for IL as shown in Table 13.

---

[1] https://github.com/irom-princeton/dppo
[2] https://github.com/ankile/robust-rearrangement

| Parameter | Value |
|---|---|
| *Optimization* | |
| Optimizer | AdamW (actor; $\eta = 1e-4$) |
| Learning rate scheduler | Cosine decay with $10\,$k warmup steps |
| Weight decay | $1e-6$ |
| Warmup steps | 1000 |
| *Training loop* | |
| Batch size | 256 |
| *Regularization & misc.* | |
| Dropout / feature noise | 0.0 |

Table 11: Hyper-parameters shared across all policy pre-training runs.

| Component | DP | Gaussian |
|---|---|---|
| *Architecture* | U-Net backbone | Feedforward MLP |
| Backbone dims | [256, 512, 1024] | [256, 256, 256] |
| Parameter count | $\approx 66\,$M | $\approx 11\,$M |
| *Horizons* | | |
| Observation $T_o$ | 1 | 1 |
| Prediction $T_p$ | 32 | 1 |
| Action $T_a$ | 8 | 8 |
| *Diffusion settings* | | – |
| DDIM Training steps | 100 | – |
| DDPM Inference steps | 4 | – |
| *Regularization* | | |
| Dropout | – | 0.1 |

Table 12: Summary of policy network configurations. DP uses U-Net based networks. Gaussian policy uses MLP based networks.

| PPO Hyperparameters | |
|---|---|
| Number of parallel environments | 1024 |
| Episode length (one_leg) | 700 |
| Episode length (lamp / round_table) | 1000 |
| Discount factor $\gamma$ | 0.999 |
| GAE $\lambda$ | 0.95 |
| Normalize advantage | true |
| Reward clipping | $\pm 5.0$ |
| Number of gradient epochs | 50 |
| Minibatches per update | 1 |
| Max grad-norm | 1.0 |
| Target KL divergence | 0.1 |
| **Residual PPO** | |
| Residual action scaling factor | 0.1 |
| Actor learning rate | $3 \times 10^{-4}$ |
| Critic learning rate | $5 \times 10^{-3}$ |
| Learning rate schedule | Cosine (warmup: 5 steps) |
| Value-loss coefficient | 1.0 |
| Entropy coefficient | 0.0 |
| Initial log-std for actor | $-1.0$ |
| **IN–RIL IL** | |
| IL update batch size | 512 |
| IL learning rate | $1 \times 10^{-4}$ |

Table 13: Hyper-parameters for online fine-tuning.

## I.2 DPPO & IDQL

In this section, we present the hyper-parameters, network structures for DPPO and IDQL experiments. We use the diffusion policy (DP) implementation from the official DPPO codebase Ren et al. (2024) for all tasks, as shown in Table 14. First, we pre-train DP separately on each Gym (Table 15) and each Robomimic (Table 16) task. Then, we fine-tune the same policy using DPPO and IDQL. Hyper-parameters for online fine-tuning are shown in Section I.2.2 and Section I.2.3. For IN–RIL fine-tuning, we use the same RL hyper-parameters, along with a fixed batch size of 256 and learning rates of 1e-4 with 1e-6 decay for all the tasks. We use UPGrad Quinton & Rey (2024) for gradient surgery, which does not need hyper-parameter tuning.

### I.2.1 DP PRE-TRAINING

| Component | Diffusion Actor (DP) | Critic (DPPO / IDQL) |
|---|---|---|
| Backbone | MLP-diffusion | MLP |
| Hidden layers | [512, 512, 512] | [256, 256, 256] |
| Condition dim | $T_o \times D_o$ | same as actor cond dim |
| Action chunk $T_a$ | 4 | — |
| Denoising steps ($K$) | 20 (pre); 10 (fine) | — |
| Time embedding dim | 16 | — |
| Predict target | $\epsilon$ | value |
| Activation | ReLU | Mish |
| Residual style | yes | - |

Table 14: Network architectures used for DPPO and IDQL experiments.

| Gym Pre-training (DP) | | | | | |
|---|---|---|---|---|---|
| Env | Obs dim | Act dim | $T_a$ | Epochs | Batch |
| Hopper-v2 | 11 | 3 | 4 | 3000 | 128 |
| Walker2D-v2 | 17 | 6 | 4 | 200 | 128 |
| HalfCheetah-v2 | 17 | 6 | 4 | 200 | 128 |
| *LR / weight decay* | LR=1e–3; weight decay=1e–6 | | | | |
| *Denoise & schedule* | denoise=20; cosine LR; 1 warmup; min LR=1e–4; EMA=0.995; save every 100 | | | | |

Table 15: Pre-training settings for the diffusion policy on Gym tasks.

| Robomimic Pre-training (DP) | | | | | |
|---|---|---|---|---|---|
| Task | Obs dim | Act dim | $T_a$ | Epochs | Batch |
| Lift | 19 | 7 | 4 | 3000 | 256 |
| Can | 23 | 7 | 1 | 3000 | 256 |
| Square | 23 | 7 | 1 | 3000 | 256 |
| Transport | 59 | 14 | 8 | 3000 | 256 |
| *LR / weight decay* | LR=1e–4; weight decay=1e–6 | | | | |
| *Denoise & schedule* | denoise=20; cosine LR; 100 warmup; min LR=1e–5; EMA=0.995; save every 500 | | | | |

Table 16: Pre-training settings for the diffusion policy on Robomimic tasks.

### I.2.2 DPPO FINE-TUNING

| Gym Fine-tuning (DPPO) | | | | | |
|---|---|---|---|---|---|
| Env | Par envs | Iter | Steps/it | Batch | Upd/it |
| All tasks | 40 | 1000 | 500 | 50 000 | 5 |
| *RL hyperparams* | $\gamma$=0.99; GAE $\lambda$=0.95; vf coeff=0.5; target KL=1.0 | | | | |
| *Learning rates* | actor=1e–4; critic=1e–3; cosine LR; 10 warmup | | | | |
| *Clipping & noise* | clip=0.2; policy-clip=0.01; randn-clip=3; min std=0.1 | | | | |
| *Denoise schedule* | discount=0.99; fine-tune $K$=10; save every 100 | | | | |

Table 17: DPPO fine-tuning settings on Gym tasks.

| Robomimic Fine-tuning (DPPO) | | | | | |
|---|---|---|---|---|---|
| Task | Par envs | Iter | Steps/it | Batch | Upd/it |
| Lift | 50 | 81 | 300 | 7 500 | 10 |
| Can | 40 | 301 | 300 | 6 000 | 10 |
| Square | 40 | 301 | 400 | 8 000 | 10 |
| Transport | 50 | 201 | 400 | 10 000 | 5 |
| *RL hyperparams* | $\gamma = 0.999$; GAE $\lambda$=0.95; vf coeff=0.5; target KL=1.0 | | | | |
| *Learning rates* | actor=1e–4; critic=1e–3; cosine LR; 10 warmup | | | | |
| *Clipping & noise* | clip=0.2; policy-clip=0.01; randn-clip=3; min std=0.1 | | | | |

Table 18: DPPO fine-tuning settings on Robomimic tasks.

### I.2.3 IDQL FINE-TUNING

| Gym Fine-tuning (IDQL) | | | | | |
|---|---|---|---|---|---|
| Env | Par envs | Iter | Steps/it | Batch | Grad/it |
| All tasks | 40 | 1000 | 500 | 1 000 | 128 |
| *RL hyperparams* | $\gamma = 0.99$; GAE $\lambda$=0.95; vf coeff=0.5; target KL=1.0 | | | | |
| *Learning rates* | actor=1e–4; critic=1e–3; cosine LR; 10 warmup | | | | |
| *Replay & batch* | buffer=25 000; batch=1 000; replay=128 | | | | |
| *Denoise schedule* | discount=0.99; fine-tune $K$=10; save every 100 | | | | |

Table 19: IDQL fine-tuning settings on Gym tasks.

| Robomimic Fine-tuning (IDQL) | | | | | |
|---|---|---|---|---|---|
| Task | Par envs | Iter | Steps/it | Batch | Grad/it |
| Lift | 50 | 120 | 300 | 1 000 | 128 |
| Can | 40 | 301 | 300 | 1 000 | 128 |
| Square | 40 | 301 | 400 | 1 000 | 128 |
| Transport | 50 | 201 | 400 | 1 000 | 128 |
| *RL hyperparams* | $\gamma = 0.999$; GAE $\lambda$=0.95; vf coeff=0.5; target KL=1.0 | | | | |
| *Learning rates* | actor=1e–4; critic=1e–3; cosine LR; 10 warmup | | | | |
| *Replay & batch* | buffer varies; batch=1 000; replay=128 | | | | |

Table 20: IDQL fine-tuning settings on Robomimic tasks.

