# OpenReview forum: "IN-RIL: Interleaved Reinforcement and Imitation Learning for Policy Fine-Tuning"
_ICLR.cc/2026/Conference — ICLR 2026 Conference Withdrawn Submission_

### Official Review · Reviewer_ohh5 · 2025-10-31

**Soundness:** 2
**Presentation:** 3
**Contribution:** 2
**Rating:** 2
**Confidence:** 4

**Summary:**

The paper introduces a method for policy finetuning by interleaving RL and IL updates.  The paper provides experimental and theoretical results showing why this approach can result in better performance.

**Strengths:**

Theoretical analysis:
The theoretical analysis is sound with appropriate assumptions and the paper includes both theoretical and empirical results.

Task selection:
The task selection makes sense and selects tasks of a variety of difficulty.

**Weaknesses:**

Novelty:
The idea of doing phases of imitation learning and reinforcement learning is not new (https://arxiv.org/abs/2206.12030), while the theoretical angle is useful, it is hard to claim novelty.

Results:
Looking at the results for most of the environments, it is hard to visually tell the method performs better than RL only. There is no mention of how many seeds the runs are over or error bars. There is also a lack of recent state of the art baselines for RL methods and imitation learning. For example, DSRL (https://arxiv.org/pdf/2506.15799) performs much better than DPPO and IDQL on robomimic but is not included.

**Questions:**

How does the proposed method perform against IBRL (https://arxiv.org/pdf/2311.02198) and DGN (https://arxiv.org/pdf/2506.07505), which both answer the question of "How to synergize the exploratory strengths of reinforcement learning with the stability of imitation learning for efficient policy fine-tuning?" that the paper mentions.

What are the experimental details such as number of seeds and evaluations? How are rewards computed?

---

### Official Review · Reviewer_wd8x · 2025-11-01

**Soundness:** 1
**Presentation:** 2
**Contribution:** 3
**Rating:** 2
**Confidence:** 4

**Summary:**

This paper addresses the problem of combining reinforcement learning (RL) and imitation learning (IL) to improve the sample efficiency and stability of RL policy fine-tuning. The key proposal is IN-RIL, a method that interleaves IL updates with RL updates, rather than combining their objectives. The paper suggests this alternating optimization allows the policy to benefit from IL's stability and RL's exploration. The authors also propose two gradient separation mechanisms (gradient surgery and network separation via a residual policy) to prevent interference between the two objectives. The method is evaluated on Robomimic, FurnitureBench, and Gym benchmarks.

**Strengths:**

- The problem of combining IL and RL for sample-efficient and stable policy fine-tuning is very important and highly relevant to robotics/RL communities.

- The authors provide a theoretical analysis to motivate the interleaving approach, and clearly stated their assumptions.

- IN-RIL is demonstrated to improve the performance of different underlying RL algorithms (e.g., DPPO, IDQL) across a variety of benchmarks.

**Weaknesses:**

- **Incomplete and Concerning Statistical Reporting:** This is a major weakness.

    - Many results reported in Tables 1 and 2 have a variability measure of ± 0.00. Several learning curves in the figures are missing error bands (e.g., Fig. 4, Walker2D for IN-RIL; Fig. 5, Mug Rack, One Leg Med). Some curves appear to be incomplete (e.g., Fig. 5, Lamp Low for the RL-only baseline). Finally, the number of seeds used and the meaning of the variability measure doesn’t appear to be stated.

    - The incomplete results and inconsistent statistical reporting gives me serious concern about the reliability of the results. It is highly unlikely that multiple trials would produce zero variance, suggesting to me that many of these results are from a single seed. This seriously weakens the empirical claims of the paper.

- **Weakly Supported Claims:**

    - The paper makes several claims about IN-RIL’s positive effect on RL exploration based only on the observation that learning curves are more stable.
    - Specifically:

         - “As shown in Figure 6(a), overly aggressive exploration in RL-only approaches can degrade performance after 0.4 × 107 steps. IN–RIL prevents this degradation… effectively constraining exploration to promising regions." (p. 9)

         - ”By periodically refreshing the agent’s memory of expert demonstrations through IL gradients, IN–RIL effectively structures exploration, enabling success on tasks that RLonly approaches cannot solve.” (p. 9)

    - This is a hypothesis, not an explanation. Direct, quantitative analysis of the policy's exploration should be provided to support this claim.

    - The double descent claim for the IL loss (Fig. 6) is not fully supported. The authors claim in the text that the IL loss had already fully converged before the RL fine-tuning, but the authors should show the IL pre-training curves to confirm this.

- In several of the experiments (e.g., Hopper, HalfCheetah, Mug-Rack), the final performance improvement of IN-RIL over the RL-only fine-tuning baseline is relatively small.

- **Minor Typo:** Line 479: "influence" should be "influenced".

**Questions:**

- The paper presents two mechanisms (gradient surgery and network separation) to reduce conflict between the IL and RL objectives. Can the authors clarify in which experiments IN-RIL uses gradient surgery and which uses network separation? Or does the IN-RIL method consist of both methods? The current presentation is not fully clear on this point.

- The theoretical analysis focuses on iteration complexity and convergence. Does this analysis provide any guarantees about the *optimality* of the final policy, especially given that the IL and RL objectives are fundamentally different?

- The key hypothesis (Fig 2) is that the IL update helps the RL update escape local optima. Have the authors considered simpler random perturbation baselines (for example, adding periodic random noise to the update gradients) to see if this alone can achieve a similar stabilizing effect?

- The gap b/w IN-RIL and RL-only (IDQL) on the Transport task (Fig 3) is very large. RL-only fails completely, while IN-RIL succeeds. The paper attributes this to guided exploration, but I don’t find this very convincing since no direct analysis of exploration was provided. Can the authors provide additional insight on this result?

---

### Official Review · Reviewer_U3Uj · 2025-11-02

**Soundness:** 3
**Presentation:** 4
**Contribution:** 3
**Rating:** 6
**Confidence:** 3

**Summary:**

This paper introduces IN-RIL (INterleaved Reinforcement and Imitation Learning), a novel fine-tuning method designed to synergize the strengths of imitation learning (IL) and reinforcement learning (RL). The core idea is to move beyond simple IL pre-training followed by RL fine-tuning or linear combinations of losses. Instead, IN-RIL periodically interleaves IL updates within the RL fine-tuning process. The authors argue this "alternating optimization" allows the policy to benefit from the stability of IL and the exploratory nature of RL, helping each objective escape the local minima of the other. To address the challenge of conflicting gradients from the two different objectives, the paper proposes "gradient separation mechanisms" to prevent destructive interference. The authors provide both theoretical analysis and strong empirical results on several robotics benchmarks, demonstrating that IN-RIL can be used as a general, algorithm-agnostic plug-in to improve sample efficiency and mitigate the performance collapse often seen in pure RL fine-tuning.

**Strengths:**

*   The concept of interleaving IL and RL updates, combined with the introduction of "gradient separation mechanisms," is a novel and interesting approach to a well-known problem. It presents a more sophisticated way to combine these learning paradigms than prior methods.
*   A significant strength of IN-RIL is its design as a general "plug-in" that is compatible with various state-of-the-art RL algorithms, both on-policy and off-policy. This makes the method broadly applicable and potentially very impactful for the community.
*   The paper does an excellent job of explicitly stating its core research question: how to synergize the exploratory strengths of RL with the stability of IL. This clarity helps frame the work and makes the motivation and contributions easy for the reader to follow.

**Weaknesses:**

*   The term "gradient separation mechanisms" is introduced as a key contribution but may be unfamiliar to many readers. Could the authors provide a more intuitive explanation of this concept? What does it mean in practice to separate the gradients, and how is it implemented (e.g., via gradient surgery or network separation)? A simple illustrative example would be very helpful.
*   The setup described in Section 2, which covers the pre-training and fine-tuning phases, is largely standard. The paper could be strengthened by more quickly moving from this background material to the novel aspects of the IN-RIL framework itself.

**Questions:**

See "Weaknesses" above

---

### Official Review · Reviewer_XhVD · 2025-11-03

**Soundness:** 2
**Presentation:** 3
**Contribution:** 2
**Rating:** 2
**Confidence:** 4

**Summary:**

The paper presents IN-RIL, an approach for policy fine-tuning that periodically alternates between imitation learning (IL) and reinforcement learning (RL) updates, rather than combining the IL and RL loss functions linearly. The authors argue that IL and RL optimize different loss landscapes with complementary local optima, and that interleaving updates enables each objective to help the other escape suboptimal solutions. Inspired by existing work, the paper uses two gradient separation mechanisms to prevent destructive interference: gradient surgery via projection and network separation using residual architectures. The paper also presents theoretical convergence analysis establishing optimal interleaving ratios and iteration complexity bounds. Experiments across Robomimic, FurnitureBench, and OpenAI Gym benchmarks demonstrate that IN-RIL improves sample efficiency and task success rates when integrated with state-of-the-art RL algorithms, including DPPO, IDQL, and residual PPO.

**Strengths:**

Originality
- The paper presents a novel non-linear approach to combining IL and RL through alternating optimization rather than linear combination, which represents a significant departure from existing regularization-based methods.
- It provides rigorous convergence analysis with Theorems 1-2 establishing optimal interleaving ratios and iteration complexity bounds under reasonable assumptions.

Quality
- Evaluation of IN-RIL spans 14 tasks across three diverse benchmarks (manipulation and locomotion, sparse and dense rewards, short and long horizons), demonstrating broad potential applicability.
- IN-RIL is shown to be compatible with multiple state-of-the-art RL algorithms (DPPO, IDQL, residual PPO), covering both on-policy and off-policy methods.

Clarity
- The paper is generally well-organized with clear motivation and effective use of figures (especially the illustration in Figure 2).

Significance
- It addresses a core problem in robot learning of fine-tuning policies using RL.​ The framework is general enough to serve as a plug-in to existing RL algorithms, enhancing its potential impact

**Weaknesses:**

Major

1. The theoretical analysis relies on strong assumptions that lack empirical validation. Specifically, Assumption 1 (gradient relationship) is never measured during training, and it remains unclear when this assumption holds in practice.

2. Additionally, Theorem 1 suggests adaptive interleaving ratios based on gradient alignment, yet all experiments use fixed ratios without justification for why the simpler approach works despite theoretical recommendations. This gap between theory and practice weakens the theoretical contribution's practical relevance.

3. The paper primarily compares against RL-only fine-tuning and behavioral cloning regularization, but lacks comparisons with other recent hybrid IL-RL methods such as DQfD [1],  Cal-QL [2], RLPD [3], TD3-BC [4], DDPGfD [5], or other similar hybrid IL-RL approaches mentioned in the related work section.

4. The hyperparameter selection for the interleaving ratio $m$ appears ad hoc. $m$ varies considerably across tasks (2 to 30 in ablations with different optima per task), but no principled selection method is provided beyond trial-and-error in the main text.

5. The paper does not adequately explain failure cases where IN-RIL provides marginal or negative benefits, such as on HalfCheetah and some other tasks. Characterizing which task properties result in IN-RIL's effectiveness would strengthen the work.

6. No information is provided on how many seeds were used to report the results in Sections 3.2 and 3.3, making it difficult to evaluate their statistical significance. Also, many of the results in Figures 4, 5, and 6 do not contain any standard deviation plots.

Minor

7. The fractional subscript notation $\theta_{t+\frac{1}{1+m(t)}}$ used in the paper is unconventional and confusing. Moreover, the subscript $t$ in $\theta_t$ is not defined. My understanding is that it refers to a cycle IL update iteration $t$ consisting of $m(t)$ RL updates between two IL updates. A more standard notation may use double subscripts like $\theta_{t,j}$ or simply number updates sequentially without the fractional indexing.

8. Some of the notations such as $c_{IL}, \sigma_{IL}, L_{RL}, N_{IL:}$ presented in line 254 are not defined in the main text.

[1] Hester, Todd, et al. "Deep q-learning from demonstrations." Proceedings of the AAAI conference on artificial intelligence. Vol. 32. No. 1. 2018.

[2] Nakamoto, Mitsuhiko, et al. "Cal-ql: Calibrated offline rl pre-training for efficient online fine-tuning." Advances in Neural Information Processing Systems 36 (2023): 62244-62269.

[3] Ball, Philip J., et al. "Efficient online reinforcement learning with offline data." International Conference on Machine Learning. PMLR, 2023.

[4] Fujimoto, Scott, and Shixiang Shane Gu. "A minimalist approach to offline reinforcement learning." Advances in neural information processing systems 34 (2021): 20132-20145.

[5] Vecerik, Mel, et al. "Leveraging demonstrations for deep reinforcement learning on robotics problems with sparse rewards." arXiv preprint arXiv:1707.08817 (2017).

**Questions:**

1. Can the authors provide empirical measurements of the gradient relationship coefficient $\rho(t)$ during training for representative tasks? How frequently are IL and RL gradients actually opposed versus aligned, and does Assumption 1 hold in practice?

2. The theory suggests an adaptive interleaving ratio ($m$) based on gradient alignment (Theorem 1), but experiments use fixed values. Have the authors implemented and tested the adaptive scheme? If not, what are the practical barriers to doing so?

3. How does IN-RIL performance scale with pre-training quality? For instance, what happens with very poor pre-training (10% of optimal IL performance) versus near-optimal pre-training, and does IN-RIL still provide benefits in the latter case?

---

### Note · Authors · 2025-11-25

I have read and agree with the venue's withdrawal policy on behalf of myself and my co-authors.